# Platonic representation of foundation machine learning interatomic potentials

**Zhenzhu Li** [1,2,3] ✉ **& Aron Walsh** [1]

Foundation machine learning interatomic potentials (MLIPs) have emerged as powerful tools for atomistic simulation, yet different models encode chemical environments in incompatible latent spaces, limiting direct comparison and interoperability. The platonic representation hypothesis suggests that sufficiently capable models converge towards a shared statistical representation of reality. Here, motivated by this hypothesis, we show that independently developed MLIPs exhibit statistically consistent geometric organization of atomic environments. By projecting embeddings relative to a set of atomic anchors, we unify the latent spaces of seven MLIPs—spanning equivariant, non-equivariant, conservative and non-conservative architectures—into a common latent space that preserves chemical periodicity and structural invariants. This unified framework enables cross-model optimal transport, interpretable embedding arithmetic and the detection of representational biases. Furthermore, we show that deviation in this space provides a ground-truth-free measure for atypical structures, and signals physical prediction failures. Our results suggest that the platonic representation offers a practical route towards interoperable, comparable and interpretable foundation models for materials science.

Machine learning interatomic potentials (MLIPs) offer a powerful tool to accelerate the inference of energy ($E$), forces ($F$) and stresses ($\sigma$) in atomistic structures. The development of MLIPs has evolved from early data-fitting approaches using crystal graphs, such as M3GNet[1], to recent architectural designs incorporating atom-centred expansions with high body-order, message passing and equivariance in implementations such as ACE, MACE, NequIP and SevenNet[2–5]. This evolution includes the divergence between conservative architectures, where forces are derived as the negative gradient of the energy, and non-conservative regimes that predict forces directly, as explored in Orb-v3[6,7]. While the Matbench Discovery[8] framework provides a benchmark for these pretrained MLIPs, current evaluations rely on metrics related to the predictive performance of the models.

Recent work in natural language processing has proposed the 'platonic representation hypothesis'. It posits that neural networks trained with different objectives and modalities converge towards shared statistical models of reality in their representation spaces[9–11]. Whether an analogous phenomenon exists in materials science remains an open question. Unlike language models that learn from human-generated text, MLIPs learn from quantum mechanical calculations that vary in fidelity (for example, choice of exchange-correlation functional, basis set) and chemical coverage. Nevertheless, the underlying physics and chemistry[12], governed by quantum mechanics and the fundamental nature of interatomic interactions, provides a ground truth that all models must approximate.

Here we demonstrate that foundation MLIPs converge towards a shared, architecture-independent latent geometry, consistent with the platonic representation hypothesis. To test this, we introduce an anchor-based projection framework that maps embeddings from any architecture into a common coordinate system, and show that it reveals statistically consistent geometric organization across seven foundation MLIPs. We then quantify the extent and limits of this convergence

[1]Department of Materials, Imperial College London, London, UK. [2]Imperial-X, Imperial College London, London, UK. [3]Imperial Global Singapore, Singapore, Singapore. ✉e-mail: zhenzhu.li@imperial.ac.uk

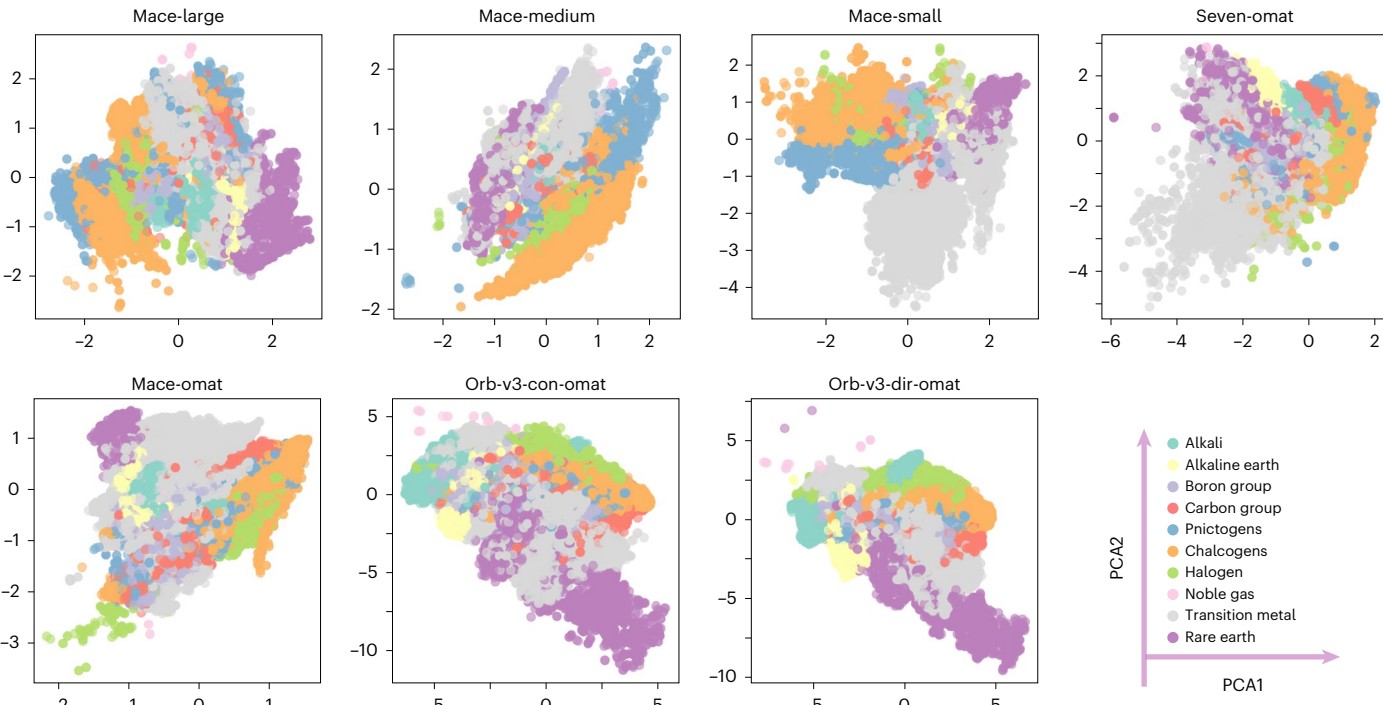

**Fig. 1 | Model-specific embeddings are incompatible before transformation.** Two-dimensional PCA projections of atomic embeddings from seven foundation MLIPs reveal distinct variance directions and element clustering patterns.

Although all models are trained to predict the same physical quantities for overlapping material sets, they learn embeddings in incompatible coordinate systems.

through complementary alignment metrics, finding substantial global agreement but persistent local divergence. To understand the origin of this shared geometry, we demonstrate that physical supervision, rather than training data distribution alone, is required for the platonic organization to emerge. Having established its physical basis, we exploit the shared geometry for cross-model embedding arithmetic and zero-shot model stitching, and show that deviations from this platonic geometry serve as a diagnostic tool for detecting architectural limitations and structural atypicality without ground-truth labels.

## Results

### Aligning incompatible representations

We chose seven foundation MLIPs representing distinct architectures, datasets and approaches to equivariance and energy conservation. These include three MACE-MP-0 variants (large, medium and small)[13] trained on the Materials Project Trajectory Dataset (MPtrj)[14]; two OMat24-based models[15] (MACE-omat and Seven-omat); and two Orb-v3 models (Orb-v3-con-omat and Orb-v3-dir-omat), which differ in their treatment of force conservation. For each model, we extracted 282,847 atomic embeddings across 27,136 structures from the MP-20 dataset[16].

We applied principal component analysis (PCA) to project these embeddings into a two-dimensional space, where the first two principal components (PCA1 and PCA2) capture the directions of greatest variance. While we also examined nonlinear visualization techniques such as uniform manifold approximation and projection (UMAP)[17] (Supplementary Fig. 1), they distort global geometry and reconfigure relative distances. As shown in Fig. 1, the raw embeddings cannot be compared directly. Dimensionality varies by architecture; for instance, MACE embeddings use 128 dimensions, whereas Orb-v3 uses 256. More importantly, the representations depend strongly on the training set-up, including the specific architecture, loss function and random initialization. Even the MACE-MP-0 variants, which share the same training dataset (MPtrj) and objective, exhibit divergent latent spaces. This indicates that original representations act in arbitrary coordinates learned by each model architecture.

To enable meaningful comparison, a unified representation is needed. We identify four desiderata for its construction: (1) model agnosticism—the construction operates on any model's latent space without access to architecture internals; (2) geometric faithfulness—relative distances and neighbourhood relationships among atomic environments are maximally preserved after projection; (3) sufficient diversity—the representation spans the chemical space broadly enough to resolve chemically distinct environments; and (4) robustness—the representation is robust to the choice of random seed and invariant to anchor permutation, ensuring that observed structure reflects intrinsic geometry rather than a particular realization. Therefore, each step of the following construction is designed to meet these criteria.

We first project the model-specific embeddings into a unified latent space. We achieve this using the relative representation strategy proposed by ref. 18, establishing a common coordinate frame independent of the original architecture—desideratum 1, model agnosticism. We use cosine similarity instead of distance functions (Euclidean $L_2$ or Manhattan $L_1$) as it provides scale invariance—critical when comparing embeddings with varying norms (for example, 3–5× differences across models; Table 1). This choice naturally bounds projections to [−1, 1], facilitating interpretation. The transformation framework is illustrated in Fig. 2a. Let $\mathbf{e}_i \in \mathbb{R}^d$ denote the original $d$-dimensional embedding of an atomic environment $i$. We select a set of $K$ anchor vectors, $\{\mathbf{a}_1, \mathbf{a}_2, ..., \mathbf{a}_K\}$, from the embedding manifold. The platonic transformation, $T(\mathbf{e}_i)$, projects $\mathbf{e}_i$ into the anchor-defined space based on its cosine similarity to these reference points:

$$\mathbf{z}_i = T(\mathbf{e}_i) = [\cos(\mathbf{e}_i, \mathbf{a}_1), \cos(\mathbf{e}_i, \mathbf{a}_2), ..., \cos(\mathbf{e}_i, \mathbf{a}_K)]^\top, \quad (1)$$

where the cosine similarity is defined as:

$$\cos(\mathbf{e}_i, \mathbf{a}_k) = \frac{\mathbf{e}_i^\top \mathbf{a}_k}{\| \mathbf{e}_i \|_2 \ \| \mathbf{a}_k \|_2}. \quad (2)$$

**Table 1 | Cross-model embedding arithmetic**

| Model | $z_{Mater}$ | $z_{Morph}$ | $z_{React}$ | $z_{React-stitch}$ |
|---|---|---|---|---|
| | ($l$, c-sim) | ($l_1$, $l_2$, i-sim, c-sim) | ($l$, c-sim) | ($l$, c-sim) |
| MACE-large | 1.53, 1.00 | 1.51, 1.59, 0.97, 1.00 | 1.26, 1.00 | 1.39, 1.00 |
| MACE-medium | 1.05, 0.84 | 1.21, 1.43, 0.95, 0.46 | 1.08, 0.69 | 1.69, 0.92 |
| MACE-small | 1.14, 0.87 | 1.62, 1.53, 0.97, 0.40 | 0.99, 0.85 | 0.99, 0.88 |
| Seven-omat | 4.50, 0.79 | 5.29, 5.50, 1.00, 0.14 | 5.58, 0.77 | 3.72, 0.38 |
| MACE-omat | 5.44, 0.79 | 5.88, 6.15, 1.00, 0.47 | 5.74, 0.75 | 4.33, 0.43 |
| Orb-v3-con | 0.68, 0.54 | 1.56, 1.83, 0.99, 0.36 | 1.05, 0.48 | 2.28, 0.82 |
| Orb-v3-dir | 1.28, 0.79 | 2.01, 2.30, 0.99, 0.39 | 1.64, 0.73 | 1.90, 0.67 |

Vector norms ($l$) and cosine similarities (c-sim, relative to MACE-large) for: (1) $Na_3MnCoNiO_6$ material embeddings; (2) $TiO_2$ polymorph differences; and (3) $BaTiO_3$ formation reaction vectors (standard versus zero-shot stitched). i-sim denotes intra-model similarity between polymorphs.

The resulting vector $z_i \in \mathbb{R}^K$ constitutes the embedding in the unified platonic space. By computing the cosine similarity between an input embedding and each anchor, we transform the absolute, model-dependent coordinates into a relative representation. The dimensionality of this new space is determined solely by the number of anchors, $K$, with each axis corresponding to the similarity to a specific anchor, rather than an arbitrary feature channel.

To ensure that the anchor set captures the diversity of the embedding manifold, we compared random sampling with Dimensionality-Reduced Encoded Clusters with Stratified (DIRECT) sampling[19]. Originally developed to facilitate MLIP training, DIRECT sampling selects points that maximize coverage in the chemical latent space. We evaluated these strategies across multiple random seeds $\in \{0, 42, 12,345\}$. As detailed in Supplementary Table 1, DIRECT sampling yields substantially more diverse anchor sets, characterized by larger pairwise distances (>1.5) and lower Silhouette scores (<0.1) compared with random selection. Consequently, we use the DIRECT strategy throughout this work to ensure broad coverage and high representational diversity (Fig. 2b).

### Convergence to a shared chemical geometry

The transformed representations obtained using anchor sets of varying sizes, $K \in \{3, 8, 20, 50, 100, 200, 400\}$, are presented in Fig. 3a. Even a minimal set of three anchors effectively aligns embeddings across disparate architectures, highlighting transition metals (grey) forming a distinct cluster, clearly separable from chalcogens (orange), halogens (green) and other main-group elements. As the anchor count increases, the unified representation stabilizes, with variance converging once the anchor set size $K$ reaches 100. Here we show relative representations from Mace-medium, Mace-small and Orb-v3-con-omat, representing both equivariant and non-equivariant architectures. A more comprehensive trend of convergence across all models is provided in Extended Data Figs. 1 and 2.

These results indicate that anchors act as stable reference points defining a shared coordinate system, satisfying requirements 2 and 4—geometric faithfulness and robustness. Once the principal geometric relationships are pinned to these anchors, the remaining embeddings naturally align, reflecting the same underlying physical drivers regardless of the model origin. Furthermore, the evolution of the embedding topology with increasing $K$ reveals a hierarchical organization within the chemical space. While a small number of anchors captures the coarse global structure, increasing the anchor density resolves finer structural details.

The quality of the unified representation also depends on anchor diversity—desideratum 3. As illustrated in Fig. 3b,c, representations constructed using DIRECT sampling yield a more unified manifold than those using random sampling. DIRECT-sampled anchors produce

tighter clustering with mean pairwise distances ranging from 1.58 to 2.67, whereas random sampling leads to skewed, loosely distributed spaces (distances of 1.80–3.11; Supplementary Table 2). This effect is particularly pronounced for models trained on the OMat24 dataset (Orb-v3 variants). Their platonic embeddings show skewed distributions relative to the MACE-MP-0 models trained on MPTrj, particularly when anchors are chosen at random. This behaviour persists across anchor set sizes from $K = 20$ to $K = 400$ (Extended Data Figs. 1 and 2).

The role of anchor diversity parallels findings in image and language representation alignment[20,21]. Small, non-diverse anchor sets (≤20) provide only a coarse global alignment; as the anchor set grows (100–400), diversity becomes essential to constrain the mapping and recover a globally consistent alignment, where the fidelity of the unified space depends on the structural richness of the probes. To validate that this geometry arises from learned physics rather than mathematical artefacts, we applied the transformation to a 'dummy' MACE model initialized with random weights (Methods). As shown in the rightmost columns of Fig. 3b,c, this untrained model yields no discernible chemical structure. This contrast confirms that the platonic representation emerges from meaningful correlations learned from the data. We further demonstrate the universality of this framework by extending it to additional architectures, including NequIP-OAM-L[22,23] and Mace-mpa-0, in Supplementary Fig. 3. All embeddings are extracted from the invariant component before energy readout, which captures the richest structural encoding after full message passing while avoiding distortion by the task-specific decoder (details in Supplementary Section 10).

### Quantifying representational interoperability

The visual convergence observed above establishes qualitative alignment; to quantitatively assess this convergence and the interoperability of embeddings in the unified space, we compute three complementary metrics at different geometric scales. Procrustes analysis ($Score_{Procrustes}$) measures global alignment: it finds the optimal rotation between two embedding distributions and reports their residual distance, quantifying whether two models organize chemical space in the same overall geometry[24]. Mutual $k$-nearest neighbours ($Score_{mKNN}$) measures local consistency: it computes the fraction of shared nearest neighbours between two models for the same atomic environments, quantifying whether two models agree on which environments are most similar. Normalized optimal transport ($Score_{OT}$) measures distributional distance: it computes the minimal effort required to morph one latent distribution into another, capturing both global and local differences[25].

As shown in Fig. 4b, foundation MLIPs show substantial global topological similarity, particularly among architectures sharing design principles (for example, MACE variants show $Score_{Procrustes} > 0.86$). Similar universal convergence is independently confirmed by ref. 26. However, this macroscale alignment masks substantial local divergence. The local neighbourhood similarity ($Score_{mKNN}$) remains low (<0.38) across all pairs (Fig. 4a), notably dropping from the values observed in the original, unaligned spaces (Supplementary Fig. 4). This disparity indicates that while different models converge on the same global physical manifold, their local neighbourhood structure remains distinct. A recent work also revealed similar divergence[27]. Notably, non-equivariant models (Orb-v3) show near-zero $Score_{mKNN}$ against equivariant models, confirming that the lack of symmetry constraints leads to a fundamentally different encoding of local atomic environments.

The optimal transport (OT) cost map (Fig. 4c) partitions the seven models into three coherent clusters, revealing the interplay between architecture and data. High OT distances separate the OMat-trained equivariant models (Seven-omat, Mace-omat) from the MACE-MP-0 family, isolating the influence of dataset composition. The largest distances, however, separate the non-equivariant Orb-v3 models from all

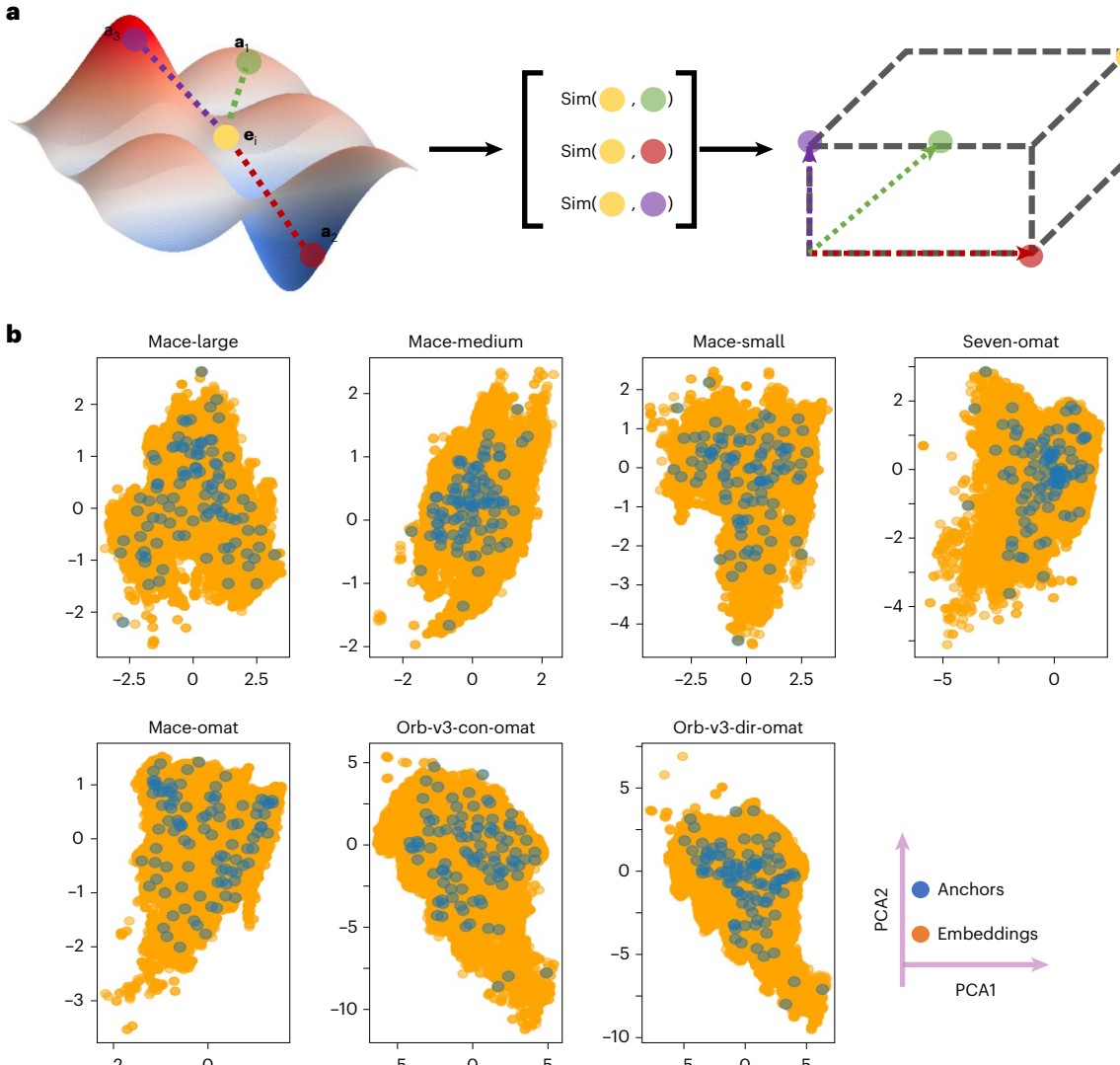

**Fig. 2 | Construction of the unified coordinate system. a**, Schematic of the anchor-based transformation. Green, red and purple points represent a set of three anchor vectors ($\mathbf{a}_k$), and the yellow point indicates a sample vector $\mathbf{e}_i \in \mathbb{R}^d$. The transformation projects $\mathbf{e}_i$ into the anchor-defined space via cosine similarity. **b**, Distribution of 100 DIRECT-sampled anchors (blue dots) overlaid on the PCA projection of the original embedding manifold (orange background).

others, confirming that disparate physical constraints create distinct energy landscapes. We can define a combined metric:

$$\text{SuperScore} = \frac{1}{2}(\text{Score}_{\text{Procrustes}} + \text{Score}_{\text{mKNN}}) \times e^{-\text{Score}_{\text{OT}}}$$

that averages the global and local alignment scores, while applying a penalty based on the transport cost. Bounded between [0, 1], Super-Score offers a summary of representational compatibility.

As an illustration, the unified representation explicitly preserves chemical structure (Fig. 4e,f; all models in Extended Data Fig. 3). When projecting element-level embeddings that are mean-pooled over all atomic environments, all models produce a similar topology of the periodic table. Elements from the same group, such as halogens, chalcogens and pnictogens, cluster coherently, and periodic trends (for example, F–Cl–Br–I) form smooth trajectories. However, architectural biases persist: equivariant models (MACE, SevenNet) produce compact, spherical clusters, whereas non-equivariant models (Orb-v3) yield skewed distributions. This suggests that without explicit symmetry constraints, the model relies more heavily on statistical correlation than physical symmetry to organize chemical space.

## Disentangling physics from data

Our results suggest that the platonic representation captures physically meaningful structure beyond what is captured by the training distribution alone (Fig. 4g–l). To isolate the role of physical supervision, we projected embeddings from Chemeleon[28]—a generative diffusion model trained on the same distribution of crystal structures and chemical compositions as the foundation MLIPs but without energy or force targets—into the unified platonic space. Despite identical data coverage, the generative model's representation lacks the tight periodic chemical organization exhibited by all foundation MLIPs (Fig. 4g,h): chemically related element groups that form distinct, well-separated clusters in the MLIP representations are heavily mixed and unresolved. This contrast demonstrates that the periodic topology of the platonic representation requires physical supervision to emerge and cannot be attributed to training data distribution alone.

To further disentangle the contributions of optimization, architecture and physics, we traced structure-level embeddings along a double-well vibrational mode of *Cmcm* SnSe, whose phonon dispersion exhibits two soft modes ($\lambda_1$, $\lambda_2$) with imaginary frequencies[29] (Fig. 4i). Structures sampled from opposite sides of the double well are related by symmetry and should map to the same point in a physically

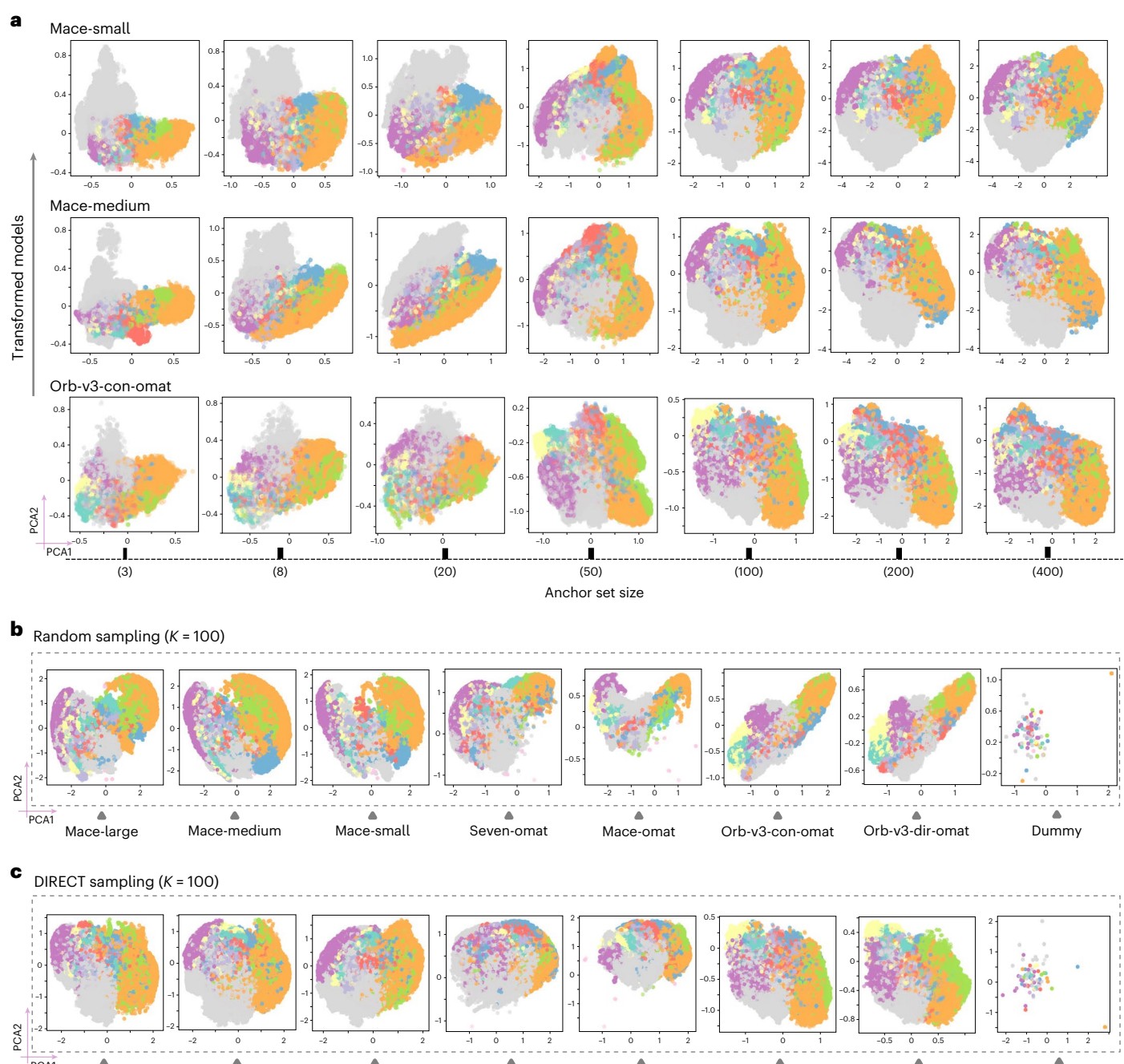

**Fig. 3 | Variation of transformed representations with anchor set size and sampling strategy. a**, Transformed representations as a function of anchor set size ($K = 3$ to $K = 400$). **b**, Two-dimensional PCA projections of converged representations using 100 randomly sampled anchors. **c**, Projections using 100 DIRECT-sampled anchors. Despite architectural diversity, all models transformed with DIRECT sampling show substantial alignment. Non-equivariant models (Orb-v3) exhibit systematic skewness. The dummy model (untrained, random weights) shows no chemical structure, confirming that alignment reflects learned physical knowledge. The colour map follows the labelling in Fig. 1.

faithful representation. The non-equivariant Orb models produce smooth but geometrically distinct embedding trajectories for the two symmetry-equivalent wells (Fig. 4j,k), indicating that these models have learned the continuity of the potential energy surface but not its symmetry. In contrast, the generative model produces a disordered, non-smooth trace (Fig. 4l), reflecting a complete absence of potential energy surface topology in its representations. These observations define a diagnostic hierarchy: a model whose embedding trace is smooth but bifurcated has internalized potential energy surface continuity but not symmetry; a model whose trace is disordered has

not learned the energy landscape at all. Both signals are accessible from the representation geometry alone.

However, a rigorous mathematical correspondence between manifold geometry and physical observables has not yet been established; we identify this as a key open problem at the intersection of machine learning and materials science.

### Algebraic consistency and model stitching
A potential utility of a platonic representation is embedding arithmetic across different models. By mapping into a unified coordinate

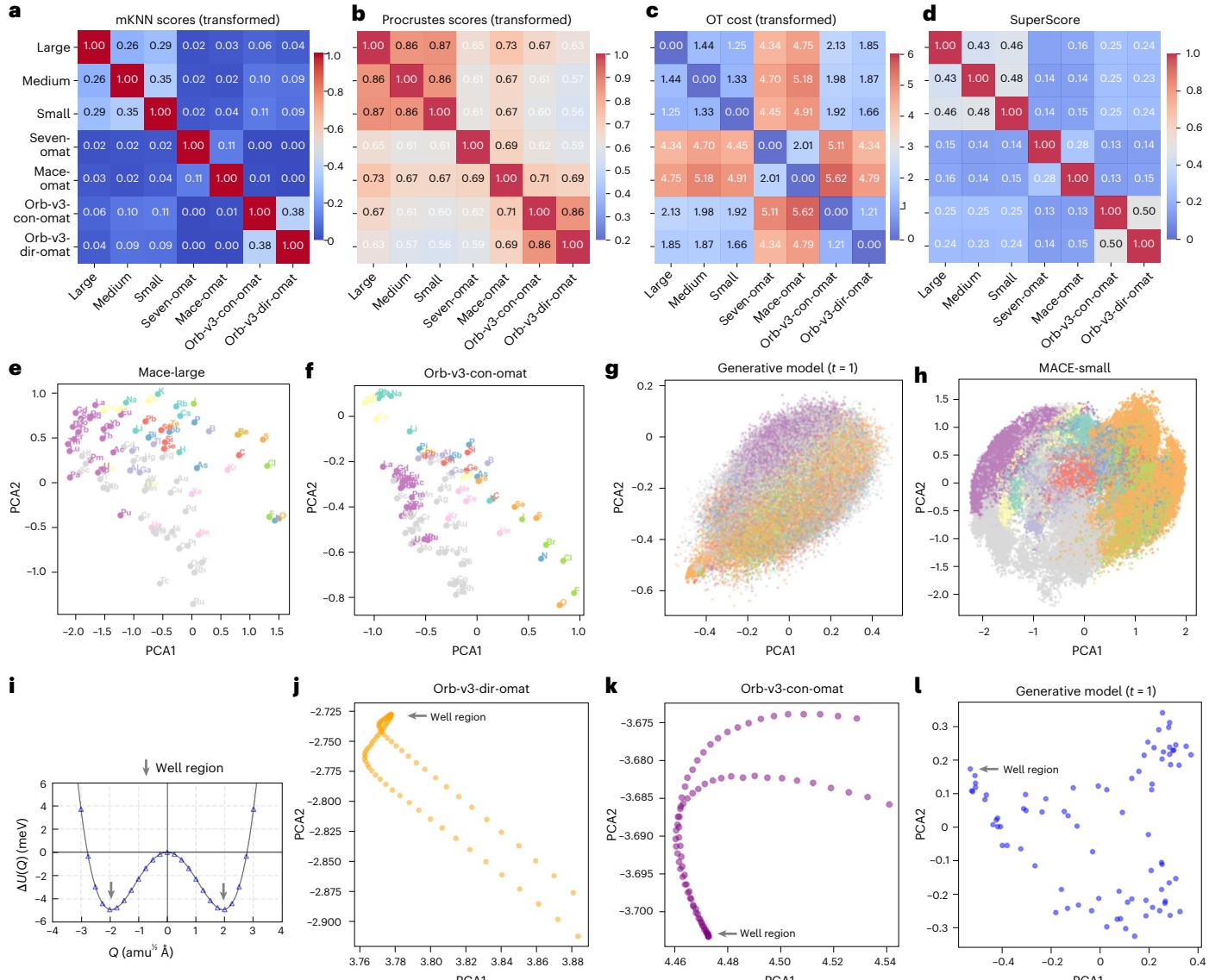

**Fig. 4 | Quantifying model similarity and chemical preservation. a**, mKNN scores (local fidelity). **b**, Procrustes scores (global alignment). **c**, Normalized OT cost. **d**, The composite SuperScore. Colour bars indicate the magnitude of each metric. Higher mKNN and Procrustes scores and lower normalized OT cost indicate better alignment and contribute to a higher composite SuperScore. **e,f**, Element-level embeddings projected into the unified space ($K = 100$ anchors) reveal consistent periodic clustering across all seven models (Extended Data Fig. 3). **g,h**, Comparison of platonic representations from a generative diffusion model (Chemeleon, $t = 1$, the best step at reconstruction) trained without energy or force supervision (**g**) and a foundation MLIP (MACE-small) (**h**). Generative model without physical constraints shows less structured latent representations

than foundation MLIP models trained with energy and force supervision. **i**, Density functional theory potential energy surface of *Cmcm* SnSe along a double-well vibrational mode with two soft modes ($\lambda_1, \lambda_2$) at imaginary frequencies; structures on either side of the well are related by symmetry. $U$, the double well potential energy; $Q$, normal mode coordinates. **j–l**, Structure-level embedding traces along this vibrational path for Orb-direct (**j**), Orb-conservative (**k**) and the generative model (**l**). Non-equivariant Orb models produce smooth but geometrically distinct branches for the two symmetry-equivalent wells. The generative model produces a disordered trace with no smooth interpolation. The colour map in **e–h** follows Fig. 1.

system, vector operations can be performed. We evaluate this through three case studies: a complex oxide ($Na_3MnCoNiO_6$), a symmetry-sensitive polymorphic pair of $TiO_2$ and a solid-state reaction to synthesize $BaTiO_3$. All embeddings are computed in the unified space ($K = 100$ anchors), with cross-model similarity (c-sim) measured relative to MACE-large and extended cases are presented in Supplementary Information.

We define the material-level embedding, $\mathbf{z}_{Mater}$, as the centroid of its atomic constituents in the unified space:

$$\mathbf{z}_{Mater} = \frac{1}{N_{atoms}} \sum_{i=1}^{N_{atoms}} \mathbf{z}_i. \tag{3}$$

As shown in Table 1, models exhibit agreement on $\mathbf{z}_{Mater}$ for $Na_3M$-$nCoNiO_6$, with c-sim values consistently between 0.79 and 0.87 (excluding Orb-v3-con-omat). Notably, OMat24-trained equivariant models (Seven-omat, Mace-omat) produce vectors with substantially larger norms ($l \approx 4.5$–$5.4$) compared with MACE-MP-0 models ($l \approx 1.1$). This scaling factor ($\sim$3–5×) is consistent across materials (Supplementary Table 3), indicating that while architectures may vary in signal magnitude, they encode similar angular information relative to the transformed axes.

To assess sensitivity to structural degrees of freedom, we analysed two $TiO_2$ polymorphs: one orthorhombic (*Pbcn*) and one tetragonal ($P4_2/mnm$) space group. While intra-model similarity (i-sim) between

polymorphs is high ($\geq 0.95$), the cross-model agreement on the difference vector, $\mathbf{z}_{Morph} = \mathbf{z}_{Mater1} - \mathbf{z}_{Mater2}$, is low ($\leq 0.40$). This suggests that while global material identity is preserved, current foundation models, and by extension their unified representations, smooth over the subtle local distortions that distinguish polymorphs, highlighting a resolution limit in current pooling strategies.

We define the reaction embedding as $\mathbf{z}_{React} = \sum \mathbf{z}_{products} - \sum \mathbf{z}_{reactants}$. For the formation of $BaTiO_3$ from its binaries, we observe consistency across models (about $\geq 0.7$ c-sim, except the Orb-v3-con model). Embedding compatibility can support zero-shot model stitching. This allows us to algebraically substitute the product state representation of one model with that of another, treating them as compatible vectors within the shared geometry. We constructed a hybrid reaction embedding, $\mathbf{z}_{React\text{-}stitch}$, by pairing reactant embeddings from MACE-large with product embeddings from other models. As detailed in Table 1, inter-model compatibility is high. MACE-MP-0 variants show strong agreement (>0.88). Surprisingly, Orb-v3-con-omat exhibits higher stitchability with MACE-large (0.82) than the other OMat24-trained models. This demonstrates that models trained on non-overlapping datasets (MPtrj versus OMat24) can be algebraically combined to yield geometrically reasonable embeddings, opening potential routes for modular reuse of pretrained foundation potentials.

### Platonic representation as a diagnostic framework

Beyond enabling cross-model algebraic operations, the shared platonic geometry also defines a physically meaningful reference frame for detecting representational deviations. We demonstrate three diagnostic applications, progressing from individual model training to architectural fidelity to physical symmetries, and structural typicality of unseen configurations.

**Tracking training dynamics.** In the context of transfer learning (Fig. 5a,b), platonic representation enables the visualization of the trajectory of atomic embeddings, distinguishing catastrophic forgetting from stable adaptation. Naive fine-tuning on Cu–Cu dimers causes unseen Au–Au embeddings to collapse (teal cross, Fig. 5a), dragging Cu weights from the correct trajectory (purple star) towards the contaminated one (teal star). Multi-head fine-tuning preserves Au knowledge (teal cross, Fig. 5b) while maintaining consistent Cu trajectories (overlapping purple and teal stars).

**Diagnosing architectural limitations.** Beyond training stability, the proposed framework also highlights fundamental limitations in architectures that do not maintain strict equivariance in their learned representations. We adopt the two-nearest-neighbour (two-NN) approach [30] to quantify effective dimension reduction arising from symmetry equivalence. Considering a reference atom (O) as an example in Fig. 5d, the ratio $d_1/d_2$—where $d_1$ and $d_2$ denote the first and second nearest-neighbour distances in representation space—serves as a metric for how the model distinguishes (or fails to distinguish) symmetry-equivalent sites. For perfectly equivariant representations, symmetry-related atoms collapse onto the same point, leading to $d_1 = 0$. By measuring the fraction of atoms with $d_1 = 0$, we estimate each model's capacity to preserve equivariant structure in its learned embeddings.

Statistics in Fig. 5c show that all equivariant MACE models maintain a consistent level of detected equivalent points across the four representative space groups studied ($P6_3/mmc$, $Fm\bar{3}m$, $I4/mmm$, $Pm\bar{3}m$), indicating robust preservation of equivariant representations. SevenNet, despite incorporating E(3)-equivariant neural network layers, does not detect $d_1 = 0$ equivariance in its original embeddings, probably owing to architectural truncation at $l_{max}$; NequIP uses stricter numerical tolerance ($1 \times 10^{-8}$ to $1 \times 10^{-10}$, float64) but still shows imperfect detection. Both recover equivariance close to MACE levels at

a tolerance of $1 \times 10^{-6}$ (Supplementary Table 6). In this regard, transformation into platonic space enables more robust detection of equivariant structure in models with numerical precision limitations, evidenced by the partial re-identification of equivalence clusters for SevenNet and NequIP models (~15–30%; Fig. 5c), suggesting that the platonic projection may reduce numerical noise.

In contrast, non-equivariant Orb models fail to identify any equivalent atoms in either their raw or platonic embeddings, deviating from physically expected symmetry relations regardless of the numerical accuracy applied, indicating that platonic transformation cannot restore equivariance that was fundamentally missing in the architecture's learning phase. This failure directly manifests in downstream predictions: as illustrated in Fig. 5d, rotating the $BaCeO_3$ structure causes the Orb-v3 embeddings to diverge rather than remain invariant, a clear instance of symmetry breaking. This violation propagates into the force field, producing qualitatively incorrect phonon dispersions (Supplementary Fig. 5).

**Ground-truth-free measure of structural deviation.** More importantly, the manifold hypothesis [31] holds that physically realistic atomic configurations concentrate near a low-dimensional manifold in representation space. Based on this, the platonic projection provides a geometry in which distances from this manifold can be interpreted as structural deviation, that is, the unified platonic space enables a ground-truth-free measure of structural typicality based on manifold distance (details in Supplementary Section 11), which is computed by (1) project an unlabelled query structure into the platonic space using the pre-computed anchor set, and (2) compute its distance and density to the reference manifold.

By partitioning the representation space into density-distance-based regions (dense/sparse interior, near/far exterior) relative to the reference manifold of known stable materials, we quantify how typical a query structure is without requiring any property labels. Relative to the near-equilibrium MP-20 anchors and the training sets of all foundation MLIPs studied, we projected configuration sampled from the non-equilibrium OMat dataset (rattled, 300 K and 1,000 K) into the platonic space (Extended Data Fig. 4c,d). Their projections remain interpretable and physically organized for both atomic-level and structural-level embeddings: 0 K relaxed structures occupy predominantly the dense interior, 300 K structures shift towards the sparse interior, and 1,000 K (Fig. 5e,f) structures shift further towards the near- and far-exterior regions. This same label-free measure extends naturally to quantifying the typicality of structures from generative models, which correlates strongly with their 'novelty' metric. In this regard, structures from GNoME[32], MatterGen (MP-20)[33] and Chemeleon are collected and their embeddings predominantly occupy the dense interior (82.8%, 78.5% and 79.6% respectively; Extended Data Fig. 4a,b), consistent with their proximity to the equilibrium training distribution. In contrast, as shown statistically in Fig. 5f, MatterGen structures conditioned on magnetic materials shift towards the sparse interior (~55% dense, ~45% sparse), reflecting configurational novelty detectable through geometry alone. We note that a rigorous mathematical proof of the correspondence between manifold geometry and physical observables does not yet exist; these results demonstrate the potential of platonic manifold distance functioning as a prospective detection measure for structural novelty. This geometric plausibility measure could provide a computationally efficient pre-screen for large-scale generative material searches, complementing real-space structural matching.

## Conclusion

We have established that foundation interatomic potentials, despite architectural heterogeneity, disjoint training sets and distinct inductive biases, exhibit a shared latent geometry that requires physical

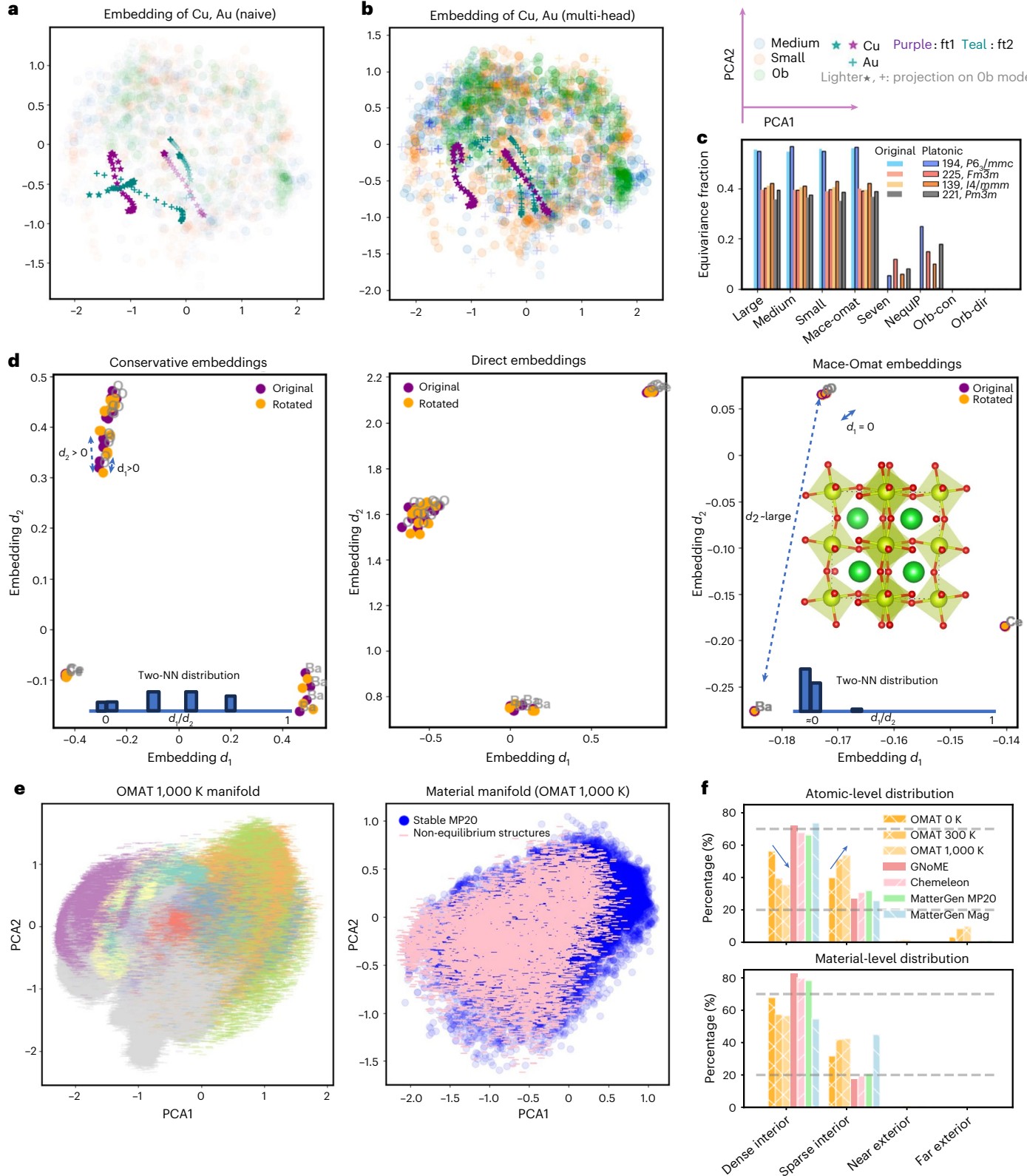

**Fig. 5 | Diagnostic applications via platonic embeddings. a,b,** Purple trajectories denote the fine-tune strategy (ft1) targeting only Cu during fine-tuning; teal trajectories represent the fine-tune strategy (ft2) targeting both Cu and Au. For both cases, only Cu–Cu dimer information is provided. **a,** Naive fine-tuning on additional Cu data causes embeddings for unseen Au atoms to collapse (teal cross (Au–Au) trajectories are artificial due to catastrophic forgetting). **b,** Multi-head fine-tuning keeps unseen Au embeddings stable near their pre-trained positions (the pair of teal cross (Au–Au) trajectories) while keeping the weights of fine-tuned Cu–Cu interactions (overlapping stared

trajectories). **c,** The equivariant fraction of embeddings from different models based on two-NN analysis. **d,** Rotational sensitivity test on $BaCeO_3$: equivariant models preserve embeddings under rotation; Orb-v3 models do not. **e,** OMAT non-equilibrium structures at 1,000 K projected into the platonic space. Left: atomic-level platonic embeddings. Right: material-level platonic embeddings (pink) compared with that of stable MP-20 structures (blue). **f,** Distribution of structures across manifold regions (dense interior, sparse interior, near exterior, far exterior) at atomic level (top) and material level (bottom).

supervision to emerge: a generative model trained on identical structural data but without energy or force targets fails to reproduce the platonic organization. This finding directly links representational convergence to the learning of physical constraints beyond statistical regularities of the training distribution. By aligning embeddings via an anchor-based projection, we unified these disparate models into a common metric space that preserves chemical periodicity, enables cross-model comparability and exposes structural invariants previously inaccessible in model-specific coordinates. We further demonstrated that the platonic geometry supports meaningful embedding arithmetic for materials, polymorphs and chemical reactions, effectively allowing zero-shot model stitching. The framework also provides a practical, ground-truth-free measure of structural typicality through manifold distance, with applications in screening generated materials and detecting out-of-distribution configurations. Simultaneously, it can signal physical prediction failures, such as symmetry breaking in non-equivariant architectures.

Beyond an immediate utility in model comparison, our results suggest a fundamental principle that as physical models in artificial intelligence for materials scale, their representations may approach compatible geometric organizations that reflect common structure learned from physical constraints. The platonic framework provides a practical mechanism to probe this convergence and a pathway to interoperable scientific models. We anticipate that future architectures may benefit from considering representational compatibility alongside performance metrics, particularly for enabling model reuse, interoperability and interpretability.

## Methods

### Definition of the platonic representation

We define the original representation of a trained machine learned potential ($M$) as the direct extraction of embeddings $e_i = \phi_i(x)$. Let $\mathcal{M} = \{M_1, M_2, \ldots, M_N\}$ be a collection of pretrained MLIPs. Each model $M_i$ defines an embedding function:

$$\phi_i : \mathcal{X} \to \mathbb{R}^d,$$

mapping atomic environments $x \in \mathcal{X}$ to latent vectors $e_i$. A canonical embedding, $z_i(x) \in \mathbb{R}^K$ is obtained by projecting each model's specific embeddings onto a model-independent anchor space using cosine similarity:

$$z_i(x) = [\cos(\phi_i(x), a_1), \cos(\phi_i(x), a_2), \ldots, \cos(\phi_i(x), a_K)]^\top,$$

where $\{a_k\}_{k=1}^K$ is an anchor set sampled from the union of model embeddings and embedded into a unified metric space. This representation satisfies four key properties:

(1) Coordinate invariance. For any orthogonal linear transformation $Q_i$ applied to the original space, $T(Q_i\phi_i(x)) \approx T(\phi_i(x))$, implying that the transformation eliminates architecture-dependent coordinate rotations.

(2) Model universality. For embeddings of the same physical environment across different models $i$ and $j$:

$$\| zi(x) - zj(x) \| \ll \| \phi i(x) - \phi j(x) \|,$$

demonstrating convergence towards a shared latent geometry.

(3) Physical latent consistency. Distances in the platonic space reflect intrinsic physical similarity:

$$\| z(x) - z(x') \| \propto \text{ similarity of atomic environments.}$$

(4) Algebraic compatibility. Structure- or reaction-level embeddings satisfy $z_{M_i}(X) \approx z_{M_j}(X)$, enabling cross-model optimal transport and arithmetic.

### DIRECT sampling strategy

To construct the anchor set, we applied the DIRECT sampling strategy[19]. This method is designed to maximize coverage of the chemical latent space. To maintain consistency, we used the distribution of embeddings from the MACE-small model as the reference manifold. Anchors were selected by applying Birch clustering to the reference distribution (target $K = 100$ clusters), followed by stratified selection to identify centroids that maximize the Silhouette score. The indices of these selected atoms were then used to retrieve the corresponding embeddings from all other models, ensuring that the anchors represent physically identical atomic environments across architectures.

### Dummy model construction

To control for architectural inductive biases, we constructed a 'dummy' baseline. This model shares the architecture of MACE-small but is initialized with random weights ($w$) and biases ($b$) drawn from a standard normal distribution. This results in a model with 3,847,696 randomized parameters that has seen no training data, serving as a negative control to verify that the platonic geometry arises from learned physical correlations rather than architectural priors.

### Similarity and distance metrics

To quantify representational alignment, we computed the following metrics.

**Procrustes score.** Measures the Euclidean distance between two representations, $A$ and $B$, after optimal orthogonal alignment. It is computed by minimizing the Frobenius norm $\|A - BQ\|_F$ over all orthogonal matrices $Q$. We report the similarity score derived from the residual sum of squares:

$$\text{Score}_{\text{Procrustes}} = 1 - \frac{\| A - BQ^* \|_F^2}{\| A - \bar{A} \|_F^2 + \| B - \bar{B} \|_F^2},$$

where $Q^*$ is the optimal rotation. A high score indicates that the two spaces share the same global topology up to rotation and scaling.

**Centred kernel alignment.** Centred kernel alignment measures the similarity between the centred Gram matrices of two representations. It captures how similarly the two spaces organize all pairwise relationships between samples, independent of the explicit rotation required by Procrustes analysis.

**OT cost.** We compute the normalized OT cost to measure the minimal effort required to transform the distribution of model $A$ into model $B$. We utilize the Sinkhorn algorithm to approximate the Wasserstein distance between the two embedding distributions in the unified Platonic space.

**Mutual $k$-nearest neighbours.** To assess local topological fidelity, we calculate the overlap of local neighbourhoods. For each sample, we identify the set of $k$-nearest neighbours in Model A and Model B. The mKNN score is defined as the Jaccard index of these two sets, averaged over all samples.

### Reporting summary

Further information on research design is available in the Nature Portfolio Reporting Summary linked to this article.

## Data availability

We utilized 27,136 structures from the MP-20 training dataset [16] as the target set for embedding extraction. While MACE models provide a native get_descriptor function, other architectures required custom interfaces to access the latent layers. An extraction script was used to generate a total of 282,847 atomic embeddings per model. All extracted

model-wise embeddings have been archived to facilitate anchor set generation and reproducibility on Zenodo at https://doi.org/10.5281/zenodo.17721681 (ref. 34).

## Code availability

All of the foundation models analysed are openly available. The codes used to perform the transformations and analysis are available in an open-source repository at https://github.com/WMD-group/Platoni-cRep (ref. 34).

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

## Acknowledgements

We thank L. Schaaf and K. Mastej for useful discussions and suggestions related to embedding analysis and the chemical consequences. We are grateful to the UK Materials and Molecular Modelling Hub for computational resources, which is partially funded by EPSRC (EP/T022213/1, EP/W032260/1 and EP/P020194/1). We thank the EPSRC for support via the AI for Chemistry: Alchemy hub (EPSRC grant EP/Y028775/1 and EP/Y028759/1) and Z.L. is funded by the Eric and Wendy Schmidt AI in Science Postdoctoral Fellowship, a Schmidt Sciences programme.

## Author contributions

The author contributions are defined following the CRediT system. Z.L.: conceptualization, investigation, formal analysis, methodology, visualization, writing—original draft. A.W.: conceptualization, methodology, writing—review and editing.

## Competing interests

A.W. is Chief Scientific Officer at CuspAI. Z.L. declares no competing interests.

## Additional information

**Extended data** is available for this paper at

**Supplementary information** The online version
contains supplementary material available at

**Correspondence and requests for materials** should be addressed to
Zhenzhu Li.

**Peer review information** *Nature Machine Intelligence* thanks the
anonymous reviewers for their contribution to the peer review of this
work. Peer reviewer reports are available.

**Publisher's note** Springer Nature remains neutral with regard to
jurisdictional claims in published maps and institutional affiliations.

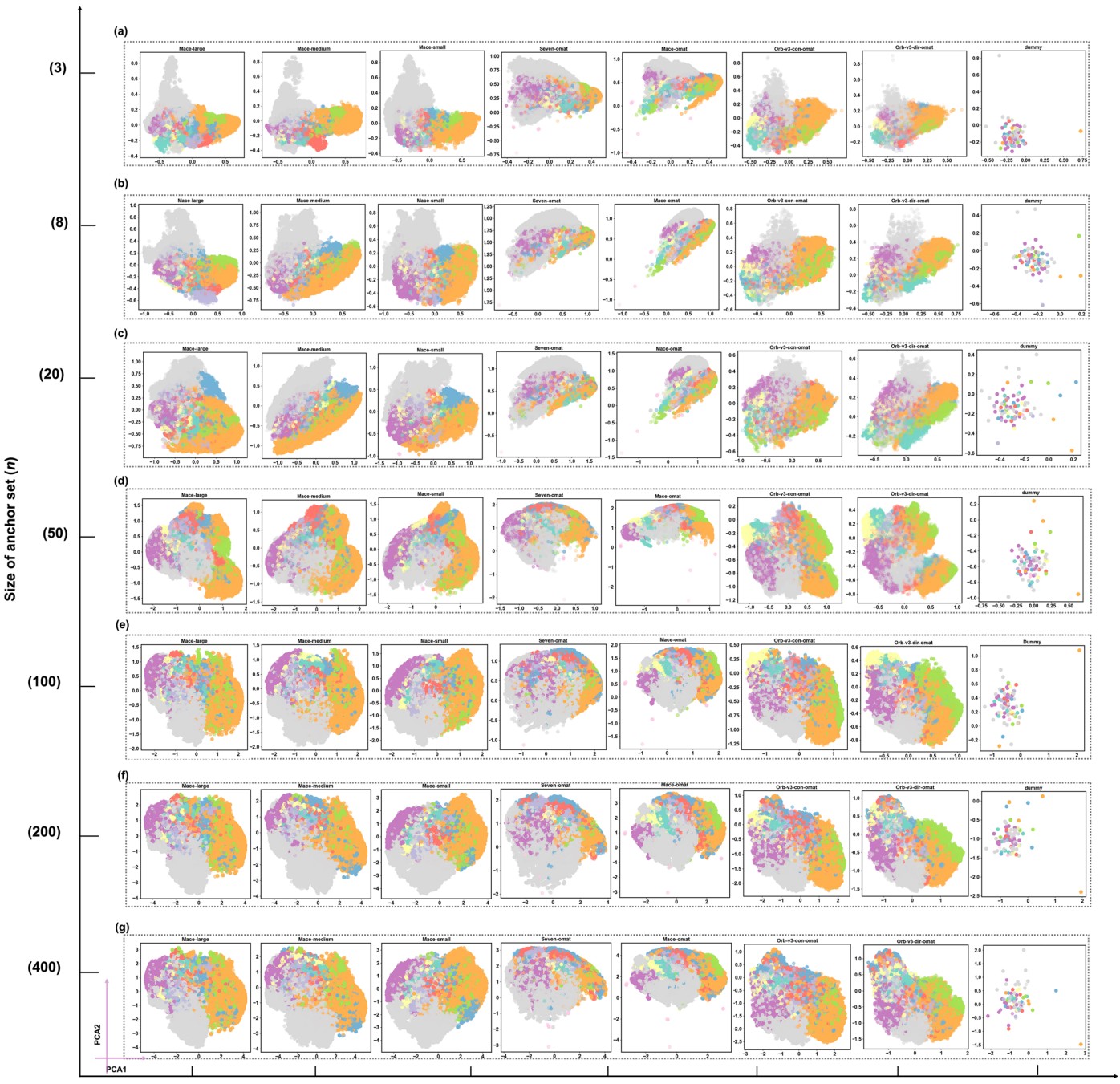

**Extended Data Fig. 1 | The Platonic transformation of seven chosen MLIPs and the dummy model corresponding to different sizes of anchor sets.** Colourmap follows Fig. 1. a–f, Platonic representations obtained with anchor set sizes of 3, 8, 20, 50, 100, 200, and 400, respectively.

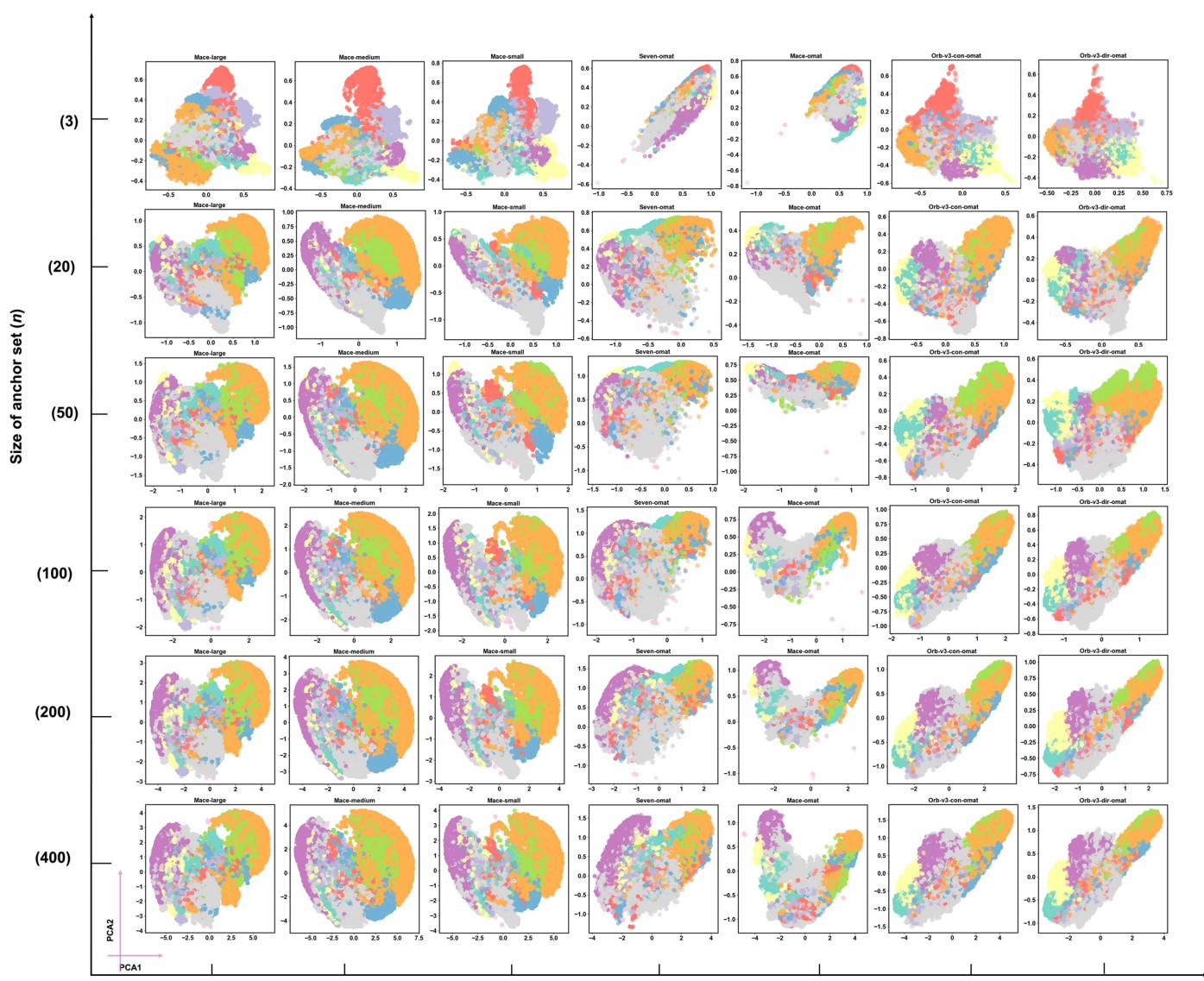

**Extended Data Fig. 2 | Seven chosen MLIPs in the unified latent space defined by different sizes of anchor sets, selected with the random sampling.** Colourmap follows Fig. 1.

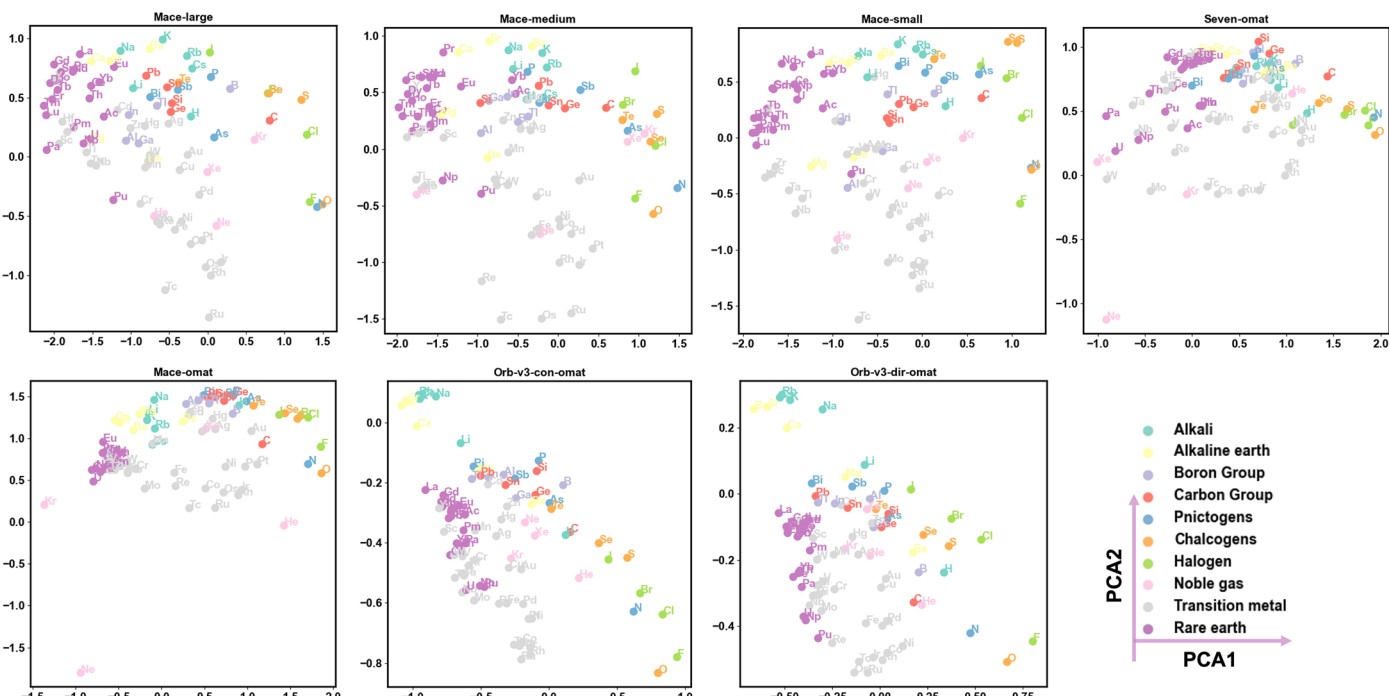

**Extended Data Fig. 3 | Element-level embeddings projected into the unified space (K=100 anchors) reveal consistent periodic clustering across all seven models.** Element-level embeddings are mean-pooled from all atomic environments in the MP-20 dataset.

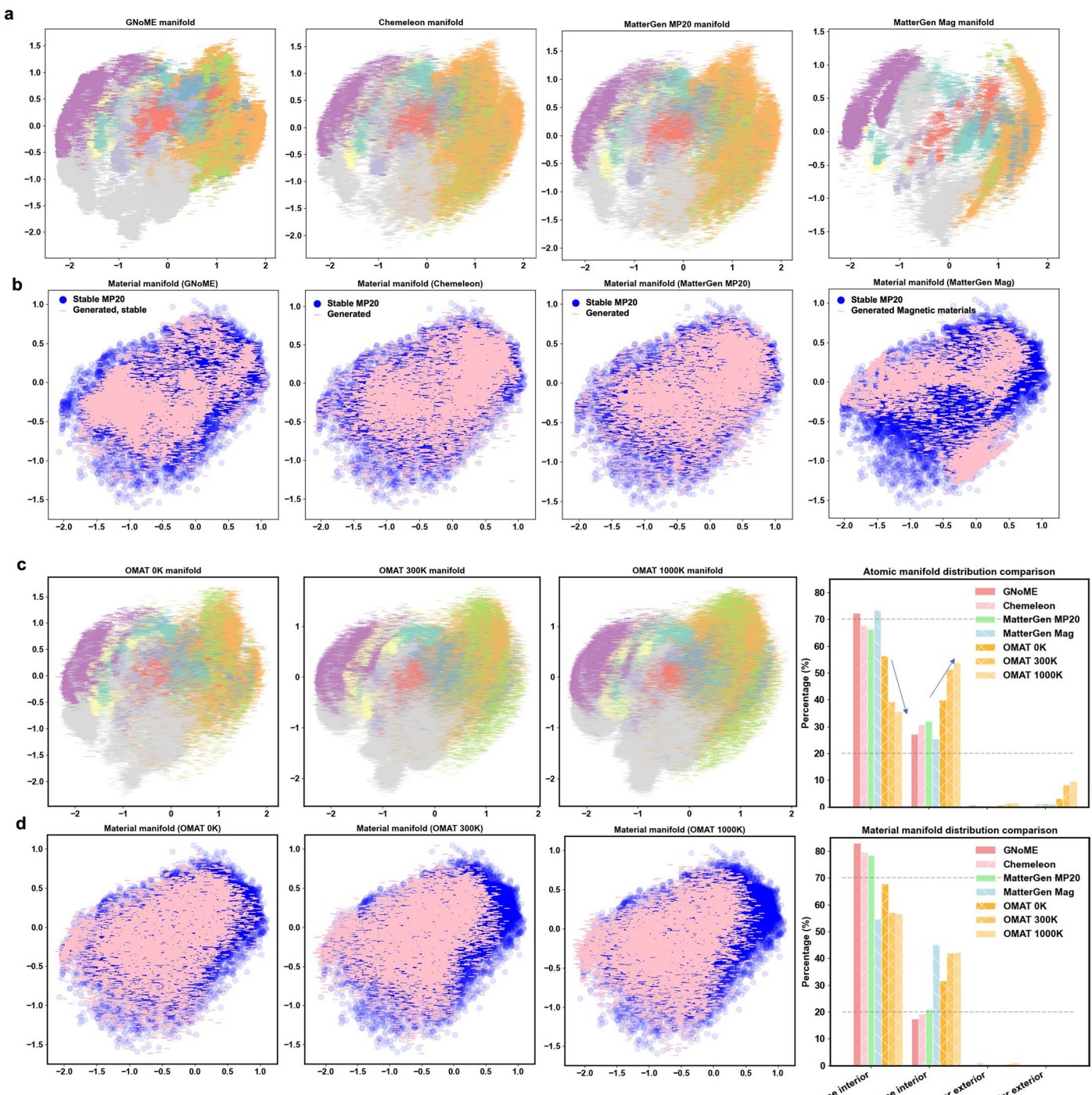

**Extended Data Fig. 4 | Platonic manifold projections of generated and non-equilibrium structures. (a, b)** Atomic-level (**a**) and material-level (**b**) Platonic embeddings of structures from GNoME, Chemeleon, and MatterGen (MP-20). (**c, d**) Atomic-level (**c**) and material-level (**d**) Platonic embeddings of OMAT non-equilibrium structures at 0 K (relaxed), 300 K, and 1000 K, projected alongside the stable MP-20 reference manifold. Colourmap of (**a,c**) follows Fig. 1.

# Reporting Summary

## Statistics

For all statistical analyses, confirm that the following items are present in the figure legend, table legend, main text, or Methods section.

| n/a | Confirmed | |
|---|---|---|
| ☐ | ☒ | The exact sample size (*n*) for each experimental group/condition, given as a discrete number and unit of measurement |
| ☐ | ☒ | A statement on whether measurements were taken from distinct samples or whether the same sample was measured repeatedly |
| ☒ | ☐ | The statistical test(s) used AND whether they are one- or two-sided *Only common tests should be described solely by name; describe more complex techniques in the Methods section.* |
| ☐ | ☒ | A description of all covariates tested |
| ☐ | ☒ | A description of any assumptions or corrections, such as tests of normality and adjustment for multiple comparisons |
| ☐ | ☒ | A full description of the statistical parameters including central tendency (e.g. means) or other basic estimates (e.g. regression coefficient) AND variation (e.g. standard deviation) or associated estimates of uncertainty (e.g. confidence intervals) |
| ☒ | ☐ | For null hypothesis testing, the test statistic (e.g. *F*, *t*, *r*) with confidence intervals, effect sizes, degrees of freedom and *P* value noted *Give P values as exact values whenever suitable.* |
| ☒ | ☐ | For Bayesian analysis, information on the choice of priors and Markov chain Monte Carlo settings |
| ☒ | ☐ | For hierarchical and complex designs, identification of the appropriate level for tests and full reporting of outcomes |
| ☒ | ☐ | Estimates of effect sizes (e.g. Cohen's *d*, Pearson's *r*), indicating how they were calculated |

*Our web collection on statistics for biologists contains articles on many of the points above.*

## Software and code

Policy information about availability of computer code

| | |
|---|---|
| Data collection | 1. Source: MP-20 dataset (publicly available, https://doi.org/10.6084/m9.figshare.25563693); 27,136 crystal structures from the Materials Project; Target set for atomic embedding extraction across all models. 2. Self-Generated Data: Atomic Embeddings (Main experimental data); Generated: 282,847 atomic embeddings per model;Models: 7 foundation MLIPs (MACE-large, MACE-medium, MACE-small, Seven-omat, MACE-omat, Orb-v3-con-omat, Orb-v3-dir-omat); Zenodo DOI: 10.5281/zenodo.17721681 (for reproducibility) 3. Anchor Sets: Self-generated via DIRECT sampling strategy 4. Control Data: self-constructed dummy model: MACE-small architecture with 3,847,696 randomized parameters (no training) 5. OMAT data, randomly selected 10000 rattled relax, 10000 at 300K, 10000 at 1000K 6. Data from generated models, 10000 each. |
| Data analysis | Machine learning foundation models for embedding extraction: MACE-MP-0a: large, medium, small; MACE-Omat-0-medium model; SevenNet-Omat; orb_v3_conservative_inf_omat; orb_v3_direct_inf_omat Anchor set selection: maml: v2025.4.1 Platonic projection: self-written code, provide in GitHub repo. Embedding similarity metrics: 1. similarity-repository, v0.1.0 2. POT: 0.9.6 Plot: self-written code using matplotlib |

For manuscripts utilizing custom algorithms or software that are central to the research but not yet described in published literature, software must be made available to editors and reviewers. We strongly encourage code deposition in a community repository (e.g. GitHub). See the Nature Portfolio guidelines for submitting code & software for further information.

## Data

Policy information about availability of data

All manuscripts must include a data availability statement. This statement should provide the following information, where applicable:

- Accession codes, unique identifiers, or web links for publicly available datasets
- A description of any restrictions on data availability
- For clinical datasets or third party data, please ensure that the statement adheres to our policy

We utilised 27,136 structures from the MP-20 training dataset as the target set for embedding extraction. While MACE models provide a native function, other architectures required custom interfaces to access the latent layers. An extraction script (https://github.com/WMD-group/PlatonicRep) was used to generate a total of 282,847 atomic embeddings per model. All extracted model-wise embeddings have been archived to facilitate anchor set generation and reproducibility in Zenodo DOI: 10.5281/zenodo.17721681.

## Research involving human participants, their data, or biological material

Policy information about studies with human participants or human data. See also policy information about sex, gender (identity/presentation), and sexual orientation and race, ethnicity and racism.

| | |
|---|---|
| Reporting on sex and gender | This study doesn't involve human sex and gender. |
| Reporting on race, ethnicity, or other socially relevant groupings | This study doesn't involve human race, ethnicity, or other socially relevant groupings. |
| Population characteristics | This study doesn't involve human population characteristics. |
| Recruitment | This study doesn't involve recruitment. |
| Ethics oversight | This study doesn't involve human ethics. |

Note that full information on the approval of the study protocol must also be provided in the manuscript.

# Field-specific reporting

Please select the one below that is the best fit for your research. If you are not sure, read the appropriate sections before making your selection.

☐ Life sciences    ☐ Behavioural & social sciences    ☒ Ecological, evolutionary & environmental sciences

For a reference copy of the document with all sections, see nature.com/documents/nr-reporting-summary-flat.pdf

# Ecological, evolutionary & environmental sciences study design

All studies must disclose on these points even when the disclosure is negative.

| | |
|---|---|
| Study description | Computational comparative analysis of latent representations across seven independently-trained machine learning interatomic potentials. Design: systematic variation of anchor set size (K = 3, 8, 20, 50, 100, 200, 400) and sampling strategy (DIRECT vs. random). Experimental units: 282,847 atomic embeddings extracted from 27,136 crystal structures (MP-20 dataset) across seven models. No biological replicates; computational reproducibility ensured via fixed random seeds {0, 42, 12345}. |
| Research sample | Research sample consists of atomic-level latent representations (embeddings) extracted from seven foundation machine learning models applied to 27,136 inorganic crystal structures from the Materials Project MP-20 dataset. Sample choice rationale: MP-20 provides broad coverage of inorganic chemical space and is a standard benchmark dataset; the seven models represent diverse architectural approaches (equivariant vs. non-equivariant, conservative vs. non-conservative) trained on overlapping but distinct datasets (MPtrj, OMat24), enabling assessment of representational convergence across model families. |
| Sampling strategy | Sample size (27,136 structures, 282,847 atoms) was determined by the complete MP-20 dataset, chosen to ensure comprehensive coverage of the inorganic chemical space represented in Materials Project data. Anchor sets were sampled using DIRECT stratified clustering to maximize chemical diversity. Anchor set size convergence was empirically validated by testing K = 3 to 400; representations stabilized at K = 100 (Figs. 3, S2-S3), confirming sufficiency. |
| Data collection | Atomic embeddings were computationally extracted from the latent layers of seven pre-trained foundation models using custom Python interfaces. For MACE models, the native get_descriptor() function was used; for other architectures (SevenNet, Orb-v3), custom extraction scripts accessed intermediate layer outputs. All extractions were performed programmatically with fixed random seeds for reproducibility. Extracted embeddings were archived in Zenodo (DOI: 10.5281/zenodo.17721681). |
| Timing and spatial scale | This is a cross-sectional computational study with no temporal sampling component. All embeddings were extracted from pre-trained model checkpoints in a single computational session (2025). Spatial scale: atomic-level representations in high-dimensional |

latent space (128-256 dimensions depending on architecture), projected into unified K-dimensional anchor space. Physical structures span the inorganic materials space represented in the Materials Project.

**Data exclusions**

No data were excluded from the analyses. All 282,847 atomic embeddings extracted from the 27,136 MP-20 structures were included in the unified representation analysis. The complete dataset ensures unbiased assessment of representational alignment across the full chemical space covered by the foundation models.

**Reproducibility**

Computational reproducibility was ensured through: (1) fixed random seeds {0, 42, 12345} for anchor sampling (Table S1); (2) version-controlled code deposited in public GitHub repository (https://github.com/WMD-group/PlatonicRep); (3) archived embeddings in Zenodo (DOI: 10.5281/zenodo.17721681); (4) documented model versions and hyperparameters. Anchor sampling was repeated across three random seeds, confirming convergence of DIRECT strategy. All computational experiments were successfully reproducible within numerical precision (tolerance 1e-6 for floating-point operations).

**Randomization**

Randomization is not applicable to this computational study. All seven foundation models were analyzed using the complete MP-20 dataset (27,136 structures) without group allocation. Anchor selection employed stratified DIRECT sampling to maximize chemical diversity rather than random allocation. Model-to-model comparisons were controlled by ensuring identical input structures across all models, eliminating structural covariates. Random seeds {0, 42, 12345} were used solely to assess sampling variance in anchor selection, not for experimental group allocation.

**Blinding**

Blinding is not relevant to this computational study. All analyses involved deterministic extraction and transformation of embeddings from pre-trained models with fixed parameters. There were no subjective assessments, human annotations, or treatment conditions that could introduce observer bias. Model identities were known throughout analysis as cross-model comparison was the explicit objective of the study. Computational reproducibility (fixed random seeds, version-controlled code) replaces the role of blinding in controlling for bias.

Did the study involve field work? ☐ Yes ☒ No

# Reporting for specific materials, systems and methods

We require information from authors about some types of materials, experimental systems and methods used in many studies. Here, indicate whether each material, system or method listed is relevant to your study. If you are not sure if a list item applies to your research, read the appropriate section before selecting a response.

## Materials & experimental systems

| n/a | Involved in the study |
|---|---|
| ☒ | Antibodies |
| ☒ | Eukaryotic cell lines |
| ☒ | Palaeontology and archaeology |
| ☒ | Animals and other organisms |
| ☒ | Clinical data |
| ☒ | Dual use research of concern |
| ☒ | Plants |

## Methods

| n/a | Involved in the study |
|---|---|
| ☒ | ChIP-seq |
| ☒ | Flow cytometry |
| ☒ | MRI-based neuroimaging |

## Plants

**Seed stocks**

This study does not involve plants, seed stocks, or biological specimens. The research analyzes computational representations (embeddings) of inorganic crystal structures from the Materials Project MP-20 dataset, which contains synthesized inorganic materials such as metal oxides, semiconductors, and ceramics. No plant material was used or collected.

**Novel plant genotypes**

This study does not involve plants or plant genotypes. No novel biological organisms were generated. The research focuses on machine learning model representations of inorganic crystalline materials.

**Authentication**

This study does not involve plants, seed stocks, or biological authentication. Model authenticity was verified through: (1) use of official pre-trained model checkpoints from published sources (MACE-MP-0, SevenNet, Orb-v3); (2) version documentation for all models; (3) validation that models reproduced expected performance on benchmark datasets as reported in original publications. The dummy control model (random weights, no training) served to verify that observed patterns arise from learned representations rather than architectural artifacts.

