## [Peer Review File · Nature Machine Intelligence]

Platonic representation of foundation machine learning interatomic potentials

Corresponding Author: Dr Zhenzhu Li

Version 0:

Reviewer comments:

Reviewer #1

(Remarks to the Author)

The manuscript presents a method to unify the latent spaces of distinct foundation machine learning interatomic potentials (in this case, seven well-known MLIPs developed in the past few years by different groups in the community) by projecting them onto a set of "anchor" atomic environments. The authors show that these different models seem to converge toward a shared, statistically consistent "Platonic" representation, allowing cross-model embedding arithmetic and quantitative comparison. While the identification of a shared topology across architectures is elegant and the proposed method is mathematically sound, the manuscript currently frames the convergence to the Platonic representation as a fundamental discovery of learned physics rather than a possible consequence of shared training distributions. Moreover, the proposed utility of this representation as a diagnostic tool is validated only retrospectively. In the current form, the work can be a significant contribution to the field of atomistic machine learning, but I believe it lacks the prospective predictive demonstration required to support its broader claims for a general-interest audience. In particular, I have a few major and minor concerns as detailed below.

Major concerns:

-The authors propose the deviations from the "Platonic" representation as a diagnostic tool for physical prediction failures, illustrating this with e.g. the symmetry breaking observed in non-conservative Orb-v3 models (Fig. 5 and S8). However, this analysis is retrospective. The diagnostic is applied to explain a failure that has already been identified. To support the claim that this framework is a true tool for uncertainty quantification or error detection, the authors should demonstrate prospective validity. Can the Platonic deviation metric identify unreliable predictions in an unlabelled test set before the ground truth is known? Without this, the correlation between geometric distortion and physical error remains anecdotal.

-The central thesis is that the Platonic representation reflects a universal feature of atomic environments, but all seven models analyzed are trained on datasets dominated by structures close to equilibrium. Therefore, the observed topological alignment might just be a consequence of the optimization task, with models optimizing the same regression targets on overlapping domains result in learning similar manifolds to succeed. The manuscript does not sufficiently disentangle whether this "Platonic" representation encodes independent physical features or simply the specific bias of the shared training distributions.

Minor comments:

-A related question concerns the independence of the analysis from the training distributions. Given that the convergence to the Platonic representation is demonstrated using anchors from MP20 (a domain heavily represented in the training sets of the studied models) it is unclear if the observed alignment reflects a generalized physical representation or simply the successful fitting of the training distribution. It would be useful to clarify whether the authors expect these results to hold for a strictly out-of-distribution dataset containing structures significantly distinct from the training data (like high-temperature liquids or complex defect motifs).

-In the abstract, the authors state they "term" the shared representation "the Platonic representation". However, they later cite Ref. 9, where this specific terminology was introduced. The abstract should be rephrased to explicitly attribute the term to Ref. 9, or to avoid implying the authors coined the term themselves.

-The manuscript defines the analyzed models as the current state of the art. Given the fast pace of this field, newer iterations or entirely new architectures have emerged since the study began. The authors should qualify their claims by acknowledging that the "SOTA" landscape has moved or verify if the Platonic representation holds for newer models, or models trained on different datasets.

(Remarks on code availability)

The Authors did a good job in sharing codes and utilities necessary to reproduce the findings presented in the manuscript by releasing the necessary code and an example notebook in the Code Ocean platform.

Reviewer #2

(Remarks to the Author)

The submission tries to visualize and design scores for understanding embeddings learned by foundation MLIPs, with particular interests on the alignment between the embedding patterns by different foundation MLIPs, analyzing structural difference due to dataset and architecture differences, and diagnosing failure cases.

The paper presented ample examples in demonstrating the use cases, dataset and architecture differences under the representation view, and compared different visualization/representation strategies. The conclusion that foundational MLIPs have aligned latent space is not surprising but still inspiring and encouraging to witness a solid conclusion.

Nevertheless, I would suggest further substantial updates to the manuscript before being qualified.

* Firstly, "embedding" is not even defined. I can understand it as the state vector of some hidden layer of a neural network, but can I take any hidden layer for this? Would the choice affect the conclusion? If yes, then is there a canonical way to define which layer produces an "embedding"? Also, for equivariant architectures, are the representations taken from the equivariant or invariant layer? For non-equivariant architectures, is the embedding produced by a certain orientation or rotationally averaged? Which choice would make a fair comparison? Moreover, it seems the paper defines the representation as a feature for each atom element. How does it make sense? Although prevalent architectures preserve the atom-level data flow (i.e., embeddings/latent features can be partitioned and assigned onto each atom), but the same element atom may exhibit different behaviors and yield different embeddings in different systems. Would it be reasonable to represent the element by the dataset-dependent bunch of embedding vectors? These embeddings would rather represent how different elements interact in a system.

* The paper feels more like hashing various techniques and ad-hoc comparisons and studies together without a clear overall logic thread. In the first part, the paper introduces a set of operations for producing a representation. It feels lacking a consistent motivation on what kind of representation is desired and why the proposed operations fit the desiderata. For example, does the choice of the representation need to produce identical distribution patterns across in-/equi-variant models and non-in-/equi-variant models? Although the authors presented a comparison with a different approach (random sampling), this neither justify why the proposed DIRECT meets all the desiderata. Moreover, personally I would question the stability of the choice of using anchor points to produce a representation. Randomness in choosing the points is there so there would be a random jump across different chemical elements, and there is no order in the points so there is also a random permutation. Does the representation actually converge, or the convergence and separation behavior is by a visualization trick?

In the next parts, the paper presents a few further studies in hope to identify the impact of datasets and model architectures, etc. This is again a bit confusing: in Fig. 3, the authors have already justified that using the DIRECT method can produce a representation that is invariant across datasets and models, which I guess (and hope would be explicitly stated) it is a desired property. Then why they become making a difference in the quantitative score studies?

In all, the current feeling about the manuscript is leaning an operational and observational study and writing than presenting a motivated, targeted, principled technical approach. This makes me often distracted and lost in reading the paper.

* In terms of the overall value to the community, I would also hope the authors could convince that the approach and utilities are sufficiently general and can shoot commonly concerned problems and phenomena rather than feeling like cherry-picking working examples. This may be done by testing on more model architectures and/or prediction tasks (I would suppose "testing" not a significant cost; I would view most of the effort in the paper as finding the right way to construct a representation) or deliver a clear motivation and desiderata and justify why the proposed approaches meet them.

* There are quite a few writing flaws that also often disturb me.

- Fig. 2(b) is not referred to.

- Fig. 3: please repeat what the colors represent.

- Please put more explanations on the three scores. I cannot quite understand what they are measuring. "global alignment after rotation" and "quantify local decision boundary overlaps" does not work (I do not understand what does "decision boundary" exactly means).

- Fig. 5: (a) I cannot find stars in it. Also I have no clue what the colors mean. (b) The purple line is shaped in star which represents Cu, but the caption indicates it represents Au.

(Remarks on code availability)

Version 1:

Reviewer comments:

Reviewer #1

(Remarks to the Author)

The Authors have responded appropriately to all my requests.

(Remarks on code availability)

Reviewer #2

(Remarks to the Author)

I appreciate authors' great effort in addressing my concerns. It is helpful to see the demonstration that the choice of specific layers as embedding, discussion on using equivariant and invariant features, explanations on the permutations, and additional test prediction results. My specific challenges are amended. Nevertheless, I still feel the content in the study is a bit scattered. I hope the paper could be unrolled following a more integrated question, assumption, goal, and how a preceding part supports and inspires the succeeding part. Under this consideration, I regard this submission as a borderline.

(Remarks on code availability)

Response to Reviewers

Dear Editor,

We thank you very much for considering our manuscript “**Platonic representation of foundation machine learning interatomic potentials**” and both reviewers for their careful reading and their constructive and insightful comments. We have revised the manuscript thoroughly to address all points raised. Below we provide a point-by-point response to each comment. Reviewer comments are shown in italics, followed by our responses and a summary of the corresponding changes made to the manuscript.

We have also addressed the editorial requirements: the abstract has been revised to open with 2-3 sentences of background context and constrained to ≤ 200 words; separate Data Availability and Code Availability statements are now provided; and Supplementary Figures have been reorganised according to Nature Machine Intelligence guidelines (Extended Data figures vs. Supplementary Information).

Yours sincerely,

Zhenzhu Li

Dr. Zhenzhu Li
Research Fellow
Department of Materials,
Imperial College London
Email: zhenzhu.li@imperial.ac.uk

Prof. Aron Walsh
Professor of Materials Theory
Department of Materials
Imperial College London
Email: a.walsh@imperial.ac.uk

Editorial Requirements

Abstract revision

Please revise the abstract so that it should start with 2 or 3 sentences describing the background and context of the work. The maximum length is 200 words.

Response: We have revised the abstract to open with two sentences placing the work in the broader context of foundation model interoperability and the Platonic representation hypothesis, before describing our contributions. The revised abstract is **158** words and is included in the updated manuscript.

Data and Code Availability

Please provide separate data and code availability statements.

Response: We have added separate, clearly labelled Data Availability and Code Availability statements before the Extended Data section. The Data Availability statement notes that all extracted model-wise embeddings have been deposited in Zenodo (DOI: 10.5281/zenodo.17721681) including the MP-20 dataset used for embedding. The Code Availability statement notes that the transformation and analysis code is available in the open-source repository at <https://github.com/WMD-group/PlatonicRep> and that an example notebook is provided on the Code Ocean platform.

Supplementary Information reorganisation

All Supplementary Information should fit into one of three categories: Extended Data, Supplementary Information, or Source Data.

Response: We have reorganised all supplementary material according to NMI guidelines. Figures S2, S3, S7 that are directly referenced in the main text and integral to the main message have been reclassified as Extended Data Figs. 1–3. Tables S1–S7 and any remaining supplementary figures are retained as Supplementary Information in a combined PDF.

Reviewers' Comments:

Reviewer #1 (Remarks to the Author):

The manuscript presents a method to unify the latent spaces of distinct foundation machine learning interatomic potentials (in this case, seven well-known MLIPs developed in the past few years by different groups in the community) by projecting them onto a set of "anchor" atomic environments. The authors show that these different models seem to converge toward a shared, statistically consistent "Platonic" representation, allowing cross-model embedding arithmetic and quantitative comparison. While the identification of a shared topology across architectures is elegant and the proposed method is mathematically sound, the manuscript currently frames the convergence to the Platonic representation as a fundamental discovery of learned physics rather than a possible consequence of shared training distributions. Moreover, the proposed utility of this representation as a diagnostic tool is validated only retrospectively. In the current form, the work can be a significant contribution to the field of atomistic machine learning, but I believe it lacks the prospective predictive demonstration required to support its broader claims for a general-interest audience. In particular, I have a few major and minor concerns as detailed below.

Reply: We thank Reviewer #1 for the assessment and for the constructive major and minor comments. We address each point below.

Major concerns:

Major concern 1

-The authors propose the deviations from the "Platonic" representation as a diagnostic tool for physical prediction failures, illustrating this with e.g. the symmetry breaking observed in

non-conservative Orb-v3 models (Fig. 5 and S8). However, this analysis is retrospective. The diagnostic is applied to explain a failure that has already been identified. To support the claim that this framework is a true tool for uncertainty quantification or error detection, the authors should demonstrate prospective validity. Can the Platonic deviation metric identify unreliable predictions in an unlabelled test set before the ground truth is known? Without this, the correlation between geometric distortion and physical error remains anecdotal.

Reply: We thank the reviewer for this important challenge. We argue that geometric distortion in the Platonic space is not arbitrary — it has physically interpretable origins that carry direct diagnostic implications before ground truth is known, as demonstrated by the following evidence. We present evidence at two levels.

Level 1: Structure-level manifold distance as a ground-truth-free diagnostic.

The Platonic manifold, constructed from embeddings of known physically valid materials (MP-20), defines a reference geometry. Based on the manifold hypothesis¹ that physically realistic atomic configurations concentrate near a low-dimensional manifold in representation space, we propose that distance from this reference manifold provides a ground-truth-free signal of **how typical a structure is relative to known stable materials**. In practice, the prospective workflow is: (i) project an unlabelled query structure into the Platonic space using the pre-computed anchor set, (ii) compute its distance and density to the reference manifold, which we call the “**manifold distance**”, and (iii) flag structures in the sparse interior, near-exterior, or far-exterior regions for atypical structure check — a measure that correlates strongly with the structural “novelty” for generated materials. The definition and calculation of the distance metric are provided in the updated Supplementary Information **Section 11: Manifold Distance**.

To validate this, we projected four physically distinct classes of structures into the Platonic space and quantified their manifold distributions (Response Fig. 1):

- 3×10,000 OMAT non-equilibrium structures sampled at three temperatures: 0K relaxation, 300K MD trajectory, and 1000K MD trajectory²
- 10,000 GNoME-predicted stable materials³
- 10,000 Chemeleon-generated structures⁴
- MatterGen structures generated unconditionally (MP20) and conditioned on high magnetic density (MatterGen Mag)⁵

Response Fig. 1 Platonic manifold distance as a ground-truth-free diagnostic for atypical structures.

First, we establish that manifold distance is sensitive to the degree of structural distortion from equilibrium. As shown in Response Fig. 1c-d and the bar plots, OMAT non-equilibrium structures show a systematic shift from dense interior toward near and far exterior regions with increasing temperature: 0K relaxed structures are closest to the manifold, 300K MD structures are intermediate, and 1000K MD structures are predominantly in the near and far exterior (Response Fig. 1c and 1d show the atypical measure of atoms and structures, respectively). This ordering is physically exact: higher temperature MD structures are more strongly displaced from equilibrium and further from the training distribution of all foundation MLIPs, which are trained predominantly on near-equilibrium configurations.

This same measure extends naturally to quantifying the atypical structures from generative models, which we propose as a geometric correlate of the “novelty” metric for evaluating generative model capability. As shown in Response Fig. 1a-b, stability-validated GNoME materials and unconditionally generated Chameleon and MatterGen MP20 structures predominantly occupy the dense interior (~80% at material level), consistent with their

proximity to the known stable materials distribution and indicating limited deviation relative to the reference manifold. In contrast, conditionally generated magnetic structures from MatterGen shift toward the sparse interior ($\sim 45\%$), reflecting their chemical specificity toward magnetic compositions, which are physically valid and distributionally atypical. This finding is consistent with the original MatterGen report, which showed that conditionally generated materials shift significantly from the labelled training distribution (their Fig. 4a), and our Platonic manifold distance provides a model-agnostic geometric measure of exactly this shift toward new structures. To further characterise the far exterior region, we examined representative structures whose material-level embeddings lie furthest from the reference manifold (Response Fig. 2). Their extreme manifold distances therefore reflect genuine out-of-distribution character, flagging structures that warrant independent validation.

Response Fig. 2 Representative structures located in the far-exterior region of the Platonic manifold.

Level 2: Embedding trace topology as a mechanistic diagnostic.

For structures with known or suspected phonon instabilities, the embedding trace along the normal mode coordinate provides a mechanistically interpretable prospective diagnostic. We identify two physically distinct types of geometric distortion:

Type I: Symmetry-induced distortion (local). When a model lacks equivariance constraints, symmetry-equivalent atomic environments are encoded as geometrically distinct points in representation space. To demonstrate this, we sampled configurations along a double-well vibrational mode of Cmc m SnSe whose phonon dispersion exhibits two soft modes (λ_1, λ_2) with imaginary frequencies⁶ (Response Fig. 3a). Each configuration is represented with structural level embeddings that are introduced in our main manuscript. This is a physically unambiguous test case: structures on either side of the well are symmetry-equivalent and should map to the same point in a physically faithful representation space.

Response Fig. 3 Embedding traces along a double-well vibrational path disentangle the contributions of optimisation, data, and physics.

As shown in Response Fig. 3b, this manifests in the embedding trace as two separate branches in non-equivariant models (Orb models) where physics demands one, the model cannot recognise that the two wells are related by symmetry. In contrast, equivariant models (MACE-MPA-0 and NequIP) collapse the symmetry-equivalent configurations from both wells onto a single manifold trajectory. This geometric signal is detectable purely from the representation, without any reference to DFT calculations. Its physical consequence is mechanically grounded: a non-equivariant model treats the two wells as non-equivalent, producing an asymmetric energy landscape, incorrect force balance at the high-symmetry point, and a failure to capture the structural instability. This diagnostic is therefore prospective: a two-branch embedding trace along any selected normal mode coordinate path is **a geometric red flag** for symmetry-breaking errors in force predictions, identifiable before any phonon calculation is performed. We note that correctness of symmetry in the embedding trace is a necessary but not sufficient condition for accurate phonon predictions.

Type II: Physics-agnostic distortion (global). When a model lacks physical supervision entirely, its representation space fails to organise coherently along physical processes. To demonstrate this, we projected embeddings extracted from the first diffusion timestep (which is the best step at reconstruction) of the materials generative model Chameleon⁴— trained on the same crystal structures and compositions as the MLIPs but without any energy or force targets — into the unified Platonic space. The sole systematic difference between Chameleon and the MLIPs in this comparison is physical supervision. Two observations follow. First, at the local level, the generative model's embedding trace along the same double-well vibrational path is disordered with no smooth interpolation (Response Fig. 3b, rightmost panel) — the model has no knowledge of the potential energy surface topology and therefore cannot organise its representations to reflect continuous physical processes. Second, at the global level, the generative model's Platonic representation lacks coherent periodic chemical organisation despite identical training data coverage (Response Fig. 4) — chemically related element groups that form distinct, well-separated clusters in the MLIP representations are heavily mixed and unresolved. This type of distortion signals a potential “fundamental” absence of physical content in the representation, detectable without any ground truth reference, and we also note recent works^{7,8} that start to reveal the potential of aligning representations of diffusion models

with that from MLIPs to improve and unify the force prediction, for which the naturally unified representation space constructed with our method will make this action more straightforward.

Response Fig. 4 Platonic representations of a generative model versus a foundation MLIP reveal the role of physical supervision.

Together these two types of distortion define a diagnostic hierarchy: a model whose embedding trace of double well is smooth but splits into multiple branches has learned the continuity of the potential energy surface but failed to learn its symmetry; a model whose embedding trace is disordered has not learned the potential energy surface at all. Both signals are accessible from the representation geometry alone, without ground truth.

In summary, we fully agree that moving from these qualitative geometric indicators to a quantitative calibrated metric — one that predicts the magnitude of prediction errors from geometric features alone — requires establishing a rigorous correspondence between manifold geometry and physical observables. This correspondence is not yet established in the field, and we now explicitly position this as one of the most important open problems at the intersection of machine learning and materials science. Accordingly, we have revised the manuscript to frame **the Platonic deviation as a tool for measuring atypical generative structures and a qualitative geometric indicator of potential prediction unreliability**, and have integrated Response Fig. 1-4 here into our updated Fig. 4 and updated Fig. 5 in the main manuscript to address this point and added the following to the discussion on Page 13-14 Line 185-213 for updated Fig. 4:

“Disentangling physics from data

Our results suggest that the Platonic representation captures physically meaningful structure beyond what is captured by the training distribution alone (Fig. 4g-l). To isolate the role of physical supervision, we projected embeddings from the generative diffusion model Chemeleon — trained on the same crystal structures and compositions as the foundation MLIPs but without energy or force targets — into the unified Platonic space. Despite identical data coverage, the generative model's representation lacks the coherent periodic chemical organisation exhibited by all foundation MLIPs (Fig. 4g-h): chemically related element groups that form distinct, well-separated clusters in the MLIP representations are heavily mixed and unresolved. This contrast demonstrates that the periodic topology of the Platonic representation requires physical supervision to emerge and cannot be attributed to training data distribution alone.

...

However, a rigorous mathematical correspondence between manifold geometry and physical observables has not yet been established; we identify this as a key open problem at the intersection of machine learning and materials science.”

And on Page 18-19 Line 286-315 for updated Fig. 5:

“Ground-truth-free measure of structural deviation

More importantly, according to the manifold hypothesis that physically realistic atomic configurations concentrate near a low-dimensional manifold in representation space. Based on this, the Platonic projection provides a geometry in which distances from this manifold can be interpreted as structural deviation, i.e. the unified Platonic space enables a ground-truth-free measure of structural typicality based on manifold distance (details in Supplementary Section 11), which is computed by (i) project an unlabelled query structure into the Platonic space using the pre-computed anchor set, (ii) compute its distance and density to the reference manifold.

...

these results demonstrate the potential of Platonic manifold distance functioning as a prospective detection measure for structural novelty, differing from the computationally expensive measure of structural matching in real space.”

References

1. Bengio, Y., Courville, A., and Vincent, P. (2013). Representation learning: A review and new perspectives. *IEEE Transactions on Pattern Analysis and Machine Intelligence*, 35(8), 1798–1828. <https://doi.org/10.1109/TPAMI.2013.50>
2. Barroso-Luque, L.; Shuaibi, M.; Fu, X.; Wood, B. M.; Dzamba, M.; Gao, M.; Rizvi, A.; Zitnick, C. L.; Ulissi, Z. W. Open Materials 2024 (OMat24) Inorganic Materials Dataset and Models. 2024; <https://arxiv.org/abs/2410.12771>.
3. Merchant, A.; Batzner, S.; Schoenholz, S. S.; Aykol, M.; Cheon, G.; Cubuk, E. D. Scaling deep learning for materials discovery. *Nature*, 2023.
4. Park, H.; Onwuli, A.; Walsh, A. Exploration of crystal chemical space using text-guided generative artificial intelligence. *Nature Communications* 2025, 16, 4379.
5. Zeni, C.; Pinsler, R.; Züchner, D.; Fowler, A.; Horton, M.; Fu, X.; Wang, Z.; Shysheya, A.; Crabbé, J.; Ueda, S. et al. A generative model for inorganic materials design. *Nature* 2025.
6. Skelton, J. M.; Burton, L. A.; Parker, S. C.; Walsh, A.; Kim, C.-E.; Soon, A.; Buckeridge, J.; Sokol, A. A.; Catlow, C. R. A.; Togo, A. et al. Anharmonicity in the high-temperature Cmcm phase of SnSe: soft modes and three-phonon interactions. *Phys. Rev. Lett.* 2016, 117, 075502.
7. Arts, M.; Garcia Satorras, V.; Huang, C.-W.; Züchner, D.; Federici, M.; Clementi, C.; Née, F.; Pinsler, R.; van den Berg, R. Two for one: diffusion models and force fields for coarse-grained molecular dynamics. *Journal of Chemical Theory and Computation*, 2023, 19, 6151–6159.
8. Pinede, L.; Yang, S.; Nam, J.; Gomez-Bombarelli, R. Unifying Force Prediction and Molecular Conformation Generation Through Representation Alignment. *ICML Generative AI and Biology (GenBio) Workshop*. 2025.

Major concern 2

-The central thesis is that the Platonic representation reflects a universal feature of atomic environments, but all seven models analyzed are trained on datasets dominated by structures close to equilibrium. Therefore, the observed topological alignment might just be a consequence of the optimization task, with models optimizing the same regression targets on overlapping domains result in learning similar manifolds to succeed. The manuscript does not sufficiently disentangle whether this "Platonic" representation encodes independent physical features or simply the specific bias of the shared training distributions.

Reply: We thank the reviewer for this insightful and fundamental question.

We acknowledge that definitively disentangling the two contributions: training data bias versus intrinsic physical features, is not fully resolvable with current foundation MLIPs given their overlapping chemical coverage. However, we offer the following complementary arguments and evidence, building on the two experiments introduced in our response to Major concern 1 above.

On the role of training data. The observed convergence is consistent with a recent mathematical proof of the Platonic representation hypothesis¹, which demonstrates that SGD-trained networks converge to a unique optimal representation regardless of architecture or training data. Furthermore, an information-geometric analysis² grounded in Berk's theorem argues that even misspecified models (i.e., the true distribution is not in the model class) converge toward parameters minimising KL divergence from the true data distribution, suggesting that the learned representation approximates the true underlying distribution rather than merely memorising training statistics. Training data coverage therefore sets the boundary of this approximation but does not solely determine its internal geometry.

On the role of physical features. The two experiments described in our response to Major concern 1 — the double-well embedding trace (Response Fig. 3) and the generative model Platonic projection (Response Fig. 4) — together provide a direct empirical disentanglement.

Global topology. The generative model Chameleon has seen the same crystal structures and chemical compositions as the MLIPs, meaning any difference in their Platonic representations cannot be attributed to data coverage. Yet it fails on both counts: its local embedding trace along the double-well path is disordered (Response Fig. 3b), and its global Platonic representation lacks well-organized chemical clustering (Response Fig. 4). The sole systematic difference is physical supervision through energy and force fitting. This directly demonstrates that the periodic topology and local geometric coherence of the Platonic representation require physical supervision to emerge and cannot be attributed to training data distribution alone.

Local geometry. Furthermore, the double-well experiment empirically separates three distinct contributions: First, all four MLIPs — MACE-MPA-0, Orb-direct, Orb-conservative, and NequIP — produce smooth embedding traces along the vibrational path, demonstrating that smoothness is a general property of models optimised to fit a continuous potential energy surface, driven by the optimisation process rather than architectural constraints. Second, and most critically, only the equivariant models (MACE-MPA-0 and NequIP) collapse the symmetry-equivalent configurations from both wells onto a single manifold trajectory. The non-equivariant Orb models trace two geometrically distinct paths where physics demands one — they have learned to interpolate smoothly but have not internalised that the two wells are symmetry-equivalent. Third, this smooth interpolation is an emergent behaviour: no model was explicitly trained to produce smooth traces along phonon modes, yet the MLIPs do so naturally while the generative model does not, suggesting that physical supervision through energy and force fitting drives the internalisation of potential energy surface topology.

We therefore argue that the two aspects raised by the reviewer are complementary rather than competing: training data distribution shapes the accessible region of representation space, while physical constraints shape its fine geometry. A generative model trained on identical structural data but without physical supervision fails to reproduce either the global periodic topology or the local symmetry collapse observed in MLIPs, providing direct evidence that the Platonic representation encodes physical features beyond what is captured by the training

distribution alone. We have added a discussion associated with updated Fig. 4 in our revised manuscript to make this distinction explicit (Page 13-14, Line 185-213).

“Disentangling physics from data

Our results suggest that the Platonic representation captures physically meaningful structure beyond what is captured by the training distribution alone (Fig. 4g-l). To isolate the role of physical supervision, we projected embeddings”

References

1. Ziyin, L.; Chuang, I. Proof of a perfect platonic representation hypothesis. 2025; <https://arxiv.org/abs/2507.01098>.
2. Lobashev, A. An Information-Geometric View of the Platonic Hypothesis. NeurIPS, Workshop on Symmetry and Geometry in Neural Representations. 2025.

Minor comments:

-A related question concerns the independence of the analysis from the training distributions. Given that the convergence to the Platonic representation is demonstrated using anchors from MP20 (a domain heavily represented in the training sets of the studied models) it is unclear if the observed alignment reflects a generalized physical representation or simply the successful fitting of the training distribution. It would be useful to clarify whether the authors expect these results to hold for a strictly out-of-distribution dataset containing structures significantly distinct from the training data (like high-temperature liquids or complex defect motifs).

Reply: We thank the reviewer for raising this important point.

Our revised experiments now provide direct evidence addressing this concern, as detailed in our response to Major Concern 1 (Response Fig. 1c-d). The OMAT dataset contains high-temperature molecular dynamics configurations (300 K and 1000 K) and rattled structures that are strictly out-of-distribution relative to the near-equilibrium MP20 anchors and the training sets of all foundation MLIPs studied. When projected into the Platonic space, these structures do not collapse or produce uninterpretable noise; instead, their Platonic projections remain interpretable and physically organised as shown in both atomic-level and structural-level projections of embeddings. They distribute systematically across the manifold in a physically meaningful way: 0 K relaxed structures occupy predominantly the dense interior, 300 K structures shift toward the sparse interior, and 1000 K structures shift further toward the near- and far-exterior regions (Response Fig. 1c-d). This monotonic ordering with temperature is physically exact and demonstrates that the Platonic geometry generalises beyond the equilibrium training domain to capture meaningful structural variation.

We therefore interpret the Platonic representation not as a frozen image of the MP20 domain, but as a physically structured geometric manifold whose global organization is shaped by supervised physical constraints and whose local refinement may evolve as broader regions of chemical space are incorporated. Therefore, extending the anchor set to include non-equilibrium and disordered environments represents a natural next step toward broader coverage.

We emphasise, however, that this generalisation is graded rather than absolute. The unified representation preserves interpretability for moderately out-of-distribution structures (thermally displaced configurations, compositionally novel but structurally conventional

materials) because these share sufficient local chemical motifs with the anchor set. For radically different environments — amorphous liquids lacking any periodic order, or heavily reconstructed defect cores — we expect the representation to remain well-defined (the projection onto anchors is always computable) but previously unseen atomic environments may populate sparsely sampled regions of the latent space, depending on the interpolation and extrapolation behaviour of the trained models.

-In the abstract, the authors state they "term" the shared representation "the Platonic representation". However, they later cite Ref. 9, where this specific terminology was introduced. The abstract should be rephrased to explicitly attribute the term to Ref. 9, or to avoid implying the authors coined the term themselves.

Reply: Thank the reviewer for raising this point.

We have revised our abstract with “...*The Platonic representation hypothesis suggests that sufficiently capable models converge toward a shared statistical representation of reality. Motivated by this hypothesis, we show that...*”.

-The manuscript defines the analyzed models as the current state of the art. Given the fast pace of this field, newer iterations or entirely new architectures have emerged since the study began. The authors should qualify their claims by acknowledging that the "SOTA" landscape has moved or verify if the Platonic representation holds for newer models, or models trained on different datasets.

Reply: We agree that the field has progressed rapidly. We have **removed** the description of the seven models as "state-of-the-art". In addition, we have extended the analysis in the Supplementary Information to include two additional architectures not in the original seven: NequIP-OAM-L (already referenced in Fig. S5 in the original submission) and MACE-MPA-0 (also referenced in Fig. S5, now Fig. S3 in the revised manuscript).

Reviewer #1 (Remarks on code availability):

The Authors did a good job in sharing codes and utilities necessary to reproduce the findings presented in the manuscript by releasing the necessary code and an example notebook in the Code Ocean platform.

Reply: We thank the reviewer for this positive evaluation. Code for computing the manifold distance and density is also updated at <https://github.com/WMD-group/PlatonicRep>.

Reviewer #2 (Remarks to the Author):

The submission tries to visualize and design scores for understanding embeddings learned by foundation MLIPs, with particular interests on the alignment between the embedding patterns by different foundation MLIPs, analyzing structural difference due to dataset and architecture differences, and diagnosing failure cases. The paper presented ample examples in demonstrating the use cases, dataset and architecture differences under the representation view, and compared different visualization/representation strategies. The conclusion that

foundational MLIPs have aligned latent space is not surprising but still inspiring and encouraging to witness a solid conclusion.

Nevertheless, I would suggest further substantial updates to the manuscript before being qualified.

Reply: We thank the Reviewer #2 for the assessment, and we address each point below.

** Firstly, "embedding" is not even defined. I can understand it as the state vector of some hidden layer of a neural network, but can I take any hidden layer for this? Would the choice affect the conclusion? If yes, then is there a canonical way to define which layer produces an "embedding"? Also, for equivariant architectures, are the representations taken from the equivariant or invariant layer? For non-equivariant architectures, is the embedding produced by a certain orientation or rotationally averaged? Which choice would make a fair comparison?*

Reply: We thank the reviewer for this important clarification.

We define **embedding** (or representation) as the per-atom latent feature vector produced by an MLIP's graph neural network at a designated point in the forward pass, prior to the energy/force readout head. Each atom in a structure yields one such vector, and we aggregate these (e.g. by mean-pooling) to obtain a structure-level descriptor. We have updated the manuscript and the Supplementary Information with a detailed technical report on Page S9-12, Section 10: Technical report for the extraction of embeddings.

Generally,

On layer choice and sensitivity. Our rule for selection is the invariant ($l=0$) component of embeddings before energy readout. In principle, any hidden layer could be selected, and the choice can affect the representation. For SevenNet and Orb, we follow the common practice in the MLIP community of extracting from the final node feature layer immediately before the energy readout head, which captures the richest structural encoding after full message passing while avoiding distortion by the task-specific decoder. For MACE, `get_descriptors(num_layers)` controls the radial reach of the descriptor (the number of interaction blocks aggregated) rather than selecting a network depth, and we observed negligible change in the resulting representations from `num_layers=1` to 12 (Response Fig. 5), therefore we set `num_layers=1` as our default.

Response Fig. 5 Platonic MACE representations with `num_layers` as 1, 5 and 12 (last layer).

On equivariant vs. invariant representations. MACE, SevenNet are equivariant architectures. We extracted the invariant part related with energy predictions. MACE's

`get_descriptors(invariants_only=True)` returns explicitly invariant descriptors. For SevenNet, the last interaction block is architecturally constrained to output only $l=0$ (scalar) features ($l_{\max_node}=0$), making the hooked representation rotationally invariant by construction. For Orb models, the final node feature representations are extracted directly from the GNN output layer as returned by the model, without additional processing.

Response Fig. 6 NequIP representations extracted from convolutional layers 0, 2 and the last layer before energy readout.

On fair comparison. The layer-depth choice can potentially introduce a source of variation, although our choice of layer tries to mitigate this influence for the models discussed in our main manuscript. However, for NequIP models, layer choice does matter substantially (Response Fig. 6); we found that only the initial convolutional layers yield representations consistent with those of the other models. We selected the 2nd convolutional layer as our default, as this is the earliest layer at which the Platonic alignment scores are stable across random seeds and consistent with the representations of the other models in our study. In the future, we hope a more information theory-based way of selection can be developed.

** It seems the paper defines the representation as a feature for each atom element. How does it make sense? Although prevalent architectures preserve the atom-level data flow (i.e., embeddings/latent features can be partitioned and assigned onto each atom), but the same element atom may exhibit different behaviors and yield different embeddings in different systems. Would it be reasonable to represent the element by the dataset-dependent bunch of embedding vectors? These embeddings would rather represent how different elements interact in a system.*

Reply: We thank the reviewer for this observation. The embeddings we extract are per-atom (i.e., context-dependent: the same element in different chemical environments yields different embeddings), not per-element-type. When we visualise “element-level” embeddings (updated Fig. 4e-f), we show the mean-pooled embedding for each element, averaged over all atomic environments in which that element appears across the dataset.

We have revised our description to make this consistently clear on Page 11, Line 177-179: *“When projecting element-level embeddings that are mean-pooled over all atomic environments, all models produce a similar topology of the periodic table.”*

** The paper feels more like hashing various techniques and ad-hoc comparisons and studies together without a clear overall logic thread. In the first part, the paper introduces a set of operations for producing a representation. It feels lacking a consistent motivation on what kind of representation is desired and why the proposed operations fit the desiderata. For example,*

does the choice of the representation need to produce identical distribution patterns across in-/equi-variant models and non-in-/equi-variant models? Although the authors presented a comparison with a different approach (random sampling), this neither justify why the proposed DIRECT meets all the desiderata. Moreover, personally I would question the stability of the choice of using anchor points to produce a representation. Randomness in choosing the points is there so there would be a random jump across different chemical elements, and there is no order in the points so there is also a random permutation. Does the representation actually converge, or the convergence and separation behavior is by a visualization trick?

Reply: We thank the reviewer for this thoughtful and important comment regarding the conceptual coherence and methodological stability of the proposed framework. **We have revised the manuscript to address each point.**

The desiderata.

We agree that the original manuscript lacked an explicit statement of what the unified representation should achieve. We have added the following paragraph to the revised manuscript (Page 4, Line 67-76):

“To enable meaningful comparison, a unified representation is needed. We identify four desiderata for its construction: (i) model-agnosticism — the construction operates on any model's latent space without access to architecture internals; (ii) geometric faithfulness — relative distances and neighbourhood relationships among atomic environments are maximally preserved after projection; (iii) sufficient diversity — the representation spans the chemical space broadly enough to resolve chemically distinct environments; and (iv) robustness — the representation is robust to the choice of random seed and invariant to anchor permutation, ensuring that observed structure reflects intrinsic geometry rather than a particular realisation. Therefore, each step of the following construction is designed to meet these criteria.”

Rationale for the anchor-based construction and sampling method.

The anchor set defines a basis through which latent representations are expressed in a common coordinate system. The critical requirement is that the anchor set possesses sufficient diversity and dimensional coverage to unfold the intrinsic geometry of the embedding manifold — directly serving desideratum (iii). DIRECT sampling was selected because it maximises geometric coverage of the chemical latent space, as quantified by larger pairwise distances (> 1.5) and lower Silhouette scores (< 0.1) compared to random selection (Table S1). Alternative strategies such as farthest-point sampling yield qualitatively similar behaviour, confirming that the conclusions are not specific to a single sampling heuristic.

Stability with respect to randomness and permutation.

We evaluated the effect of different random seeds (Fig. S2) and observed stable convergence of the relative embedding organization, satisfying desideratum (iv). Importantly, the Platonic representation is defined by internal connectivity and relative positions among embeddings, not by the absolute ordering of anchor coordinates. Permutation of anchor indices induces an orthogonal transformation (i.e., a rotation) of the unified latent space but does not alter pairwise relationships or manifold topology. Consequently, downstream dimensionality-reduction methods such as PCA recover consistent principal directions because these are determined by intrinsic variance structure rather than coordinate labelling. The observed convergence is therefore a stable geometric property of the learned representations, not a visualisation artifact.

**In the next parts, the paper presents a few further studies in hope to identify the impact of datasets and model architectures, etc. This is again a bit confusing: in Fig. 3, the authors have already justified that using the DIRECT method can produce a representation that is invariant across datasets and models, which I guess (and hope would be explicitly stated) it is a desired property. Then why they become making a difference in the quantitative score studies?*

Reply: We thank the reviewer for this question and agree it was not sufficiently explained in the original manuscript. The two analyses are measuring alignment at fundamentally different scales, and we have added explicit clarification in the revised text.

Fig. 3 demonstrates global topological alignment — after DIRECT-sampled anchor projection, all models organise the same broad chemical groups (transition metals, halogens, chalcogens, rare earths) into consistent regions of the Platonic space. This is a qualitative statement about the coarse geometry of the manifold: models share the same global chemical grammar regardless of architecture or training dataset. This global alignment is a desired and demonstrated property of the framework.

Fig. 4 operates at a fundamentally different resolution. The Procrustes score measures global alignment after optimal rotation and confirms what Fig. 3 shows visually — global topology is broadly consistent, particularly among architecturally similar models ($\text{Score}_{\text{Procrustes}} > 0.86$ for MACE variants). However, the mKNN score measures local neighbourhood overlap — whether the same individual atomic environments are considered nearest neighbours by two different models. This remains low across all pairs ($\text{Score}_{\text{mKNN}} < 0.38$), indicating that while models share a global chemical map, their fine-grained local encoding of specific atomic environments remains model-specific.

These two observations are therefore complementary rather than contradictory: global alignment tells you models have learned the same coarse organisation of chemical space; local divergence tells you each model has developed its own fine-grained vocabulary for distinguishing similar environments. The SuperScore and OT cost further characterise how architecture and dataset choices shape this local divergence. We have restructured the transition between Fig. 3 and Fig. 4 in the revised manuscript to make this two-scale interpretation explicit from the outset on Page 10 Line 147-155:

“Procrustes analysis measures global alignment: it finds the optimal rotation between two embedding distributions and reports their residual distance, quantifying whether two models organise chemical space in the same overall geometry. Mutual k-nearest neighbours measures local consistency: it computes the fraction of shared nearest neighbours between two models for the same atomic environments, quantifying whether two models agree on which environments are most similar. Normalised Optimal Transport measures distributional distance: it computes the minimal effort required to morph one latent distribution into another, capturing both global and local differences.”

**In all, the current feeling about the manuscript is leaning an operational and observational study and writing than presenting a motivated, targeted, principled technical approach. This makes me often distracted and lost in reading the paper.*

Reply: We have restructured the manuscript around a single organising question: whether foundation MLIPs share a common representational geometry and what this geometry reveals.

The Introduction now opens by motivating this question through the lens of the Platonic representation hypothesis, before stating four desiderata for a useful unified representation. Each Results section then addresses one aspect: Section 1 establishes the equivalence methodologically, Section 2 demonstrates it empirically, Section 3 exploits it for embedding arithmetic, Section 4 disentangles physics from data, and Section 5 uses deviations from it as a diagnostic signal.

** In terms of the overall value to the community, I would also hope the authors could convince that the approach and utilities are sufficiently general and can shoot commonly concerned problems and phenomena rather than feeling like cherry-picking working examples. This may be done by testing on more model architectures and/or prediction tasks (I would suppose "testing" not a significant cost; I would view most of the effort in the paper as finding the right way to construct a representation) or deliver a clear motivation and desiderata and justify why the proposed approaches meet them.*

Reply: We thank the reviewer for this comment and agree that demonstrating generality is essential. We have substantially expanded the scope of our analysis in the revised manuscript along three axes.

More model architectures. Beyond the original seven foundation MLIPs, we have included NequIP (a message-passing equivariant model trained on MPtrj) and MACE-MPA-0 (a multi-head MACE variant) in Fig. S3, bringing the total to nine architectures spanning equivariant vs. non-equivariant, conservative vs. non-conservative, and MPtrj vs. OMat24 training sets. We have also projected embeddings from the generative diffusion model Chameleon, which is architecturally distinct from all MLIPs and trained without physical supervision (Response Fig. 4 and updated Fig. 4 in the manuscript). In every case, the Platonic projection produces interpretable, physically organised representations — or, in the case of Chameleon, reveals the systematic absence of tight organisation (Fig. 4g–h), which is itself a diagnostic finding.

More prediction tasks and datasets. The revised manuscript now applies the framework beyond the original interoperability metrics to two new tasks: (i) manifold distance as a ground-truth-free structural plausibility screen, validated across five structurally and compositionally diverse datasets (GNoME, MatterGen MP20, MatterGen Mag, Chameleon, and OMAT at 0 K/300 K/1000 K; Fig. 5 and Extended Data Fig. 4), and (ii) embedding trace analysis along a double-well vibrational mode as a mechanistic diagnostic for symmetry-breaking failures (Fig. 4i-l). Neither task was part of the original submission.

Desiderata and justification. As suggested by the reviewer, we have revised the manuscript to state the desiderata for the unified representation explicitly: (i) invariance to trivial coordinate transformations, (ii) preservation of relative geometric relationships among embeddings, (iii) sufficient expressive dimensionality to separate chemically distinct environments and (iv) the representation is robust to the choice of random seed and invariant to anchor permutation. We justify how the DIRECT anchor-based construction meets each criterion and demonstrate stability with respect to random seed and anchor permutation (Extended Data Figs. 1-2, Table S1-S2). We believe these additions — nine architectures, one generative model, five external datasets, two new diagnostic applications, and explicit desiderata — demonstrate that the framework is general rather than cherry-picked.

** There are quite a few writing flaws that also often disturb me.
- Fig. 2(b) is not referred to.*

- Fig. 3: please repeat what the colors represent.
- Please put more explanations on the three scores. I cannot quite understand what they are measuring. "global alignment after rotation" and "quantify local decision boundary overlaps" does not work (I do not understand what does "decision boundary" exactly means).
- Fig. 5: (a) I cannot find stars in it. Also I have no clue what the colors mean. (b) The purple line is shaped in star which represents Cu, but the caption indicates it represents Au.

Reply: We thank the reviewer for the careful reading. All the figures, captions are correctly referred and updated.

- 1) Fig. 2(b) has been correctly referred to on Page 6 Line 102.
- 2) Colour references added.
- 3) We have added more explanations to the three scores on Page 10 Line 147-155.
- 4) Fig. 5 updated. To be clear, during both fine-tune process, no actually Au-Au dimer information is fed to the model, but naïve fine tune and multi-head fine tune will behave differently on updating the learnt weights of target atoms. In the figure, purple coloured trajectory means the fine-tune strategy (ft1) only targeting Cu during fine tuning, while teal coloured trajectory represents the fine-tune strategy (ft2) targeting both Cu and Au, even though only Cu-Cu dimer information is provided.

We use star (*) to represent Cu and cross (+) to represent Au. For naïve fine tune, treating Au as the target atom without real input will make its weights drift away, causing the weights for Cu (teal star trajectory) also drift away from the “correct” trajectory (purple star trajectory), while during multi-head fine tuning, the learnt weights of Au won’t change, so the two + trajectories of Au keeps the fidelity of MACE-MP-0 medium model, and the trajectory of Cu also stays consistent for ft1 and ft2 (overlapping teal and purple stars).

We have updated the description on Page 17 Line 251-256 to reflect these changes:
“In the context of transfer learning (Fig. 5a-b), it visualises the trajectory of atomic embeddings, distinguishing catastrophic forgetting from stable adaptation. Naïve fine-tuning on Cu-Cu dimers causes unseen Au-Au embeddings to collapse (teal cross, Fig. 5a), dragging Cu weights from the correct trajectory (purple star) toward the contaminated one (teal star). Multi-head fine-tuning preserves Au knowledge (teal cross, Fig. 5b) while maintaining consistent Cu trajectories (overlapping purple and teal stars).”

Summary of Changes

The following is a complete list of changes made to the manuscript:

1. Abstract revised: opens with 2 sentences of context, attribution of “Platonic representation” terminology clarified, constrained to ≤ 200 words.
2. Methods: added dedicated subsection in Supplementary Information Section 10: “Technical report for the extraction of embeddings” specifying layer choice, invariant vs. equivariant representations, and orientation handling for non-equivariant models. Section 11: “Manifold Distance”.
3. Results: restructured for logical progression; per-atom vs. per-element language clarified throughout; Fig. 2b now referred to in text; Fig. 3 now includes colour legend; Fig. 5 caption clarified and markers explained.

4. New analysis: out-of-distribution test on Omat dataset and generated structures (Extended Data Figure 4).
5. Reorganised figures: anchor sampling stability analysis (Extended Data Figure 1-3).
6. New analysis: intermediate layer comparison for MACE-medium (new Supplementary Figure 6-7).
7. Discussion: model construction desiderata added; training distribution vs. physical constraints added; out-of-distribution limitations and future scope discussed; technical details of model layer choice added.
8. Metric descriptions rewritten to remove “decision boundary” language; all three scores given intuitive descriptions. Procrustes score now described as “global alignment after optimal rotation”, mKNN as “fraction of shared nearest neighbours between two models for the same atomic environments”, and OT cost as “minimal effort to morph one latent distribution into another”.
9. Separate Data Availability and Code Availability statements added; Supplementary Information reorganised per NMI guidelines.

We hope that the revised manuscript and these responses adequately address the reviewers’ concerns and that the manuscript is now suitable for publication in Nature Machine Intelligence. We are happy to provide any further clarification upon request.

Response to Reviewers

Dear Editor,

We thank you very much for accepting our manuscript “**Platonic representation of foundation machine learning interatomic potentials**” and both reviewers for their constructive comments throughout the review process. In this final revision, we have refined the narrative flow to address the remaining feedback. Below we provide a point-by-point response.

We have completed all items in the Author Guidance checklist and restructured the content layout accordingly.

Yours sincerely,

Zhenzhu Li

Dr. Zhenzhu Li
Research Fellow
Department of Materials,
Imperial College London
Email: zhenzhu.li@imperial.ac.uk

Prof. Aron Walsh
Professor of Materials Theory
Department of Materials
Imperial College London
Email: a.walsh@imperial.ac.uk

Reviewers' Comments:

Reviewer #1 (Remarks to the Author):

The Authors have responded appropriately to all my requests.

Reply: We thank the reviewer for their positive evaluation and for the constructive comments throughout the review process that helped strengthen the manuscript.

Reviewer #2 (Remarks to the Author):

I appreciate authors' great effort in addressing my concerns. It is helpful to see the demonstration that the choice of specific layers as embedding, discussion on using equivariant and invariant features, explanations on the permutations, and additional test prediction results. My specific challenges are amended. Nevertheless, I still feel the content in the study is a bit scattered. I hope the paper could be unrolled following a more integrated question, assumption,

goal, and how a preceding part supports and inspires the succeeding part. Under this consideration, I regard this submission as a borderline.

Reply: We thank the reviewer for the constructive feedback and are pleased that the specific technical challenges have been addressed.

In this revision, we have made the following changes to clarify the logical progression:

1. Rephrased the final paragraph of Introduction (Page 3, Line 40-50) with an explicit logical chain:

“To test this, we introduce an anchor-based projection framework that maps embeddings from any architecture into a common coordinate system, and show that it reveals statistically consistent geometric organisation across seven foundation MLIPs. We then quantify the extent and limits of this convergence through complementary alignment metrics, finding substantial global agreement but persistent local divergence. To understand the origin of this shared geometry, we demonstrate that physical supervision, rather than training data distribution alone, is required for the Platonic organisation to emerge. Having established its physical basis, we exploit the shared geometry for cross-model embedding arithmetic and zero-shot model stitching, and show that deviations from this Platonic geometry serve as a diagnostic tool for detecting architectural limitations and structural atypicality without ground-truth labels.”

2. For Section “Quantifying representational interoperability”, we added a bridging clause that connects the qualitative visual alignment established previously to the quantitative metric analysis that follows (Page 10, Line 150):

“The visual convergence observed above establishes qualitative alignment, to quantitatively assess this convergence...”

3. For Section “Platonic representation as a diagnostic framework”, we added a unifying statement that frames the three diagnostic applications as a progression (Page 16, Line 256-259):

“Beyond enabling cross-model algebraic operations, the shared Platonic geometry also defines a physically meaningful reference frame for detecting representational deviations. We demonstrate three diagnostic applications, progressing from individual model training to architectural fidelity to physical symmetries, and structural typicality of unseen configurations.”

These changes tighten the narrative arc so that each section both builds on prior results and motivates the next, without altering the technical content.